# Modelling Ocean Colour Derived Chlorophyll-a

Stephanie Dutkiewicz[1,2], Anna E. Hickman[3], Oliver Jahn[1]

[1]Department of Earth, Atmospheric and Planetary Sciences, Massachusetts Institute of Technology, Cambridge, 02139 MA, USA

[2]Center for Climate Change Science, Massachusetts Institute of Technology, Cambridge, 02139 MA, USA

[3]Ocean and Earth Sciences, University of Southampton, National Oceanography Centre Southampton, Southampton, SO14 3ZH, United Kingdom.

*Correspondence to*: Stephanie Dutkiewicz (stephd@mit.edu)

**Abstract.** This article provides a proof-of-concept for using a biogeochemical/ecosystem/optical model with radiative transfer

component as a laboratory to explore aspects of ocean colour. We focus here on the satellite ocean colour Chlorophyll-a (Chl-a) product provided by the often-used blue/green reflectance ratio algorithm. The model produces output that can be compared directly to the real world ocean colour remotely sensed reflectance. This model output can then be used to produce an ocean colour satellite-like Chl-a product using an algorithm linking the blue versus green reflectance similar to that used for the real world. Given that the model includes complete knowledge of the (model) water constituents, optics and reflectance, we can

explore uncertainties and their causes in this proxy for Chl-a (called "derived Chl-a" in this paper). We compare the derived Chl-a to the "actual" model Chl-a field. In the model we find that the mean absolute bias due to the algorithm is 22% between derived and actual Chl-a. The real world algorithm is found using concurrent in situ measurement of Chl-a and radiometry. We ask whether increased in situ measurements to train the algorithm would improve the algorithm, and find a mixed result. There is a global overall improvement, but at the expense of some regions, especially in lower latitudes where the biases

increase. Not surprisingly, we find that regional specific algorithms provide a significant improvement, at least in the annual mean. However, in the model, we find that no matter how the algorithm coefficients are found there can be a temporal mismatch between the derived Chl-a and the actual Chl-a. These mismatches stem from temporal decoupling between Chl-a and other optically important water constituents (such as coloured dissolved organic matter and detrital matter). The degree of decoupling differs regionally and over time. For example, in many highly seasonal regions, the timing of initiation and peak of the spring

bloom in the derived Chl-a lags the actual Chl-a by days and sometimes weeks. These results indicate care should also be taken when studying phenology through satellite derived products of Chl-a. This study also re-emphasises that ocean colour derived Chl-a is not the same as the real in situ Chl-a. In fact the model derived Chl-a compares better to real world satellite-derived Chl-a than the model actual Chl-a. Modellers should keep this is mind when evaluating model output with ocean colour Chl-a and in particular when assimilating this product. Our study spans several disciplines: Our goal is to illustrate the use of

numerical laboratory that a) helps users of ocean colour, particularly modellers, gain further understanding of the products they use; and b) the ocean colour community could use to explore other ocean colour products, their biases and uncertainties, as well as to aid in future algorithm development.

## 1 Introduction

Satellite ocean colour measurements have allowed the scientific community an unprecedented ability to study phytoplankton on a global scale and at regular and frequent intervals. In particular, ocean colour products have been used extensively to explore seasonal and interannual variability, trends in ocean surface chlorophyll-a (Chl-a), and in climate, biogeochemical and ecological model evaluation. And yet, there remains a large degree of uncertainty in the satellite-derived Chl-a, with estimates ranging from 30% to >50% (Moore et al., 2009). Uncertainties arise from clouds, patchiness, atmospheric correction, measurement errors, as well as the algorithm used to deduce Chl-a. Here we focus on the uncertainty from one of these algorithms.

The most commonly used algorithm for estimating Chl-a from ocean colour uses the fact that phytoplankton absorb more in the blue range of the light spectrum than the green (Morel and Prieur, 1977). The ratio of amount of blue to green light reflected at the ocean surface at any location therefore supplies information on the concentration of Chl-a. Using datasets of coincident radiometric observations and in situ Chl-a, a 4th order polynomial can be constructed to estimate Chl-a from measured blue/green reflectance ratios (e.g. O'Reilly et al, 2000). This empirical algorithm is then used globally with satellite remotely sensed reflectance. The relationship is typically considered robust in open ocean conditions where the optical effects of phytoplankton co-vary with other optical constituents, including coloured dissolved organic matter (CDOM) and detritus, so called Case-I conditions (Smith and Baker 1978, Morel 1988, O'Reilly et al, 2000). Though even in these waters the error estimate is about 35% (Moore et al., 2009). Uncertainties that arise from the type of algorithm can be attributed to the potential divergence in the relative role of the optically important water constituents (see e.g. Siegel et al 2005b; Brown et al., 2008). There is significant ongoing work to improve algorithms. For instance, the newest National Aeronautics and Space Administration (NASA) reprocessing of Chl-a products has included a merged approach that uses different combination of reflectance bands at low Chl-a (Hu et al., 2012). There have also been many attempts to develop more mechanistically derived algorithms,using for instance known relationships between absorption, scattering and reflectance (e.g. Werdell et al., 2013a). Here we focus on the Chl-a estimated from the blue/green reflectance as it is still the most commonly known product, and until very recently used in products downloaded from both NASA and the European Space Agency (ESA) data portals, as well as merged products such as the Ocean Colour Climate Change Initiative (OC-CCI). However we note that similar techniques used in this paper could help inform on other algorithms.

That the satellite-derived products have large errors and specific regional biases is relatively well understood in the ocean colour scientific community (Hu et al., 2000, Moore et al., 2009; Blondeau-Parissier et al., 2014; Szeto et al., 2011). However, there remain many aspects of errors, biases and uncertainties that are poorly quantified, particularly in regions where there are little or no in situ data to compare to the satellite derived products. Further, many users of ocean colour products whose main expertise are in other arenas (e.g. biogeochemical and ecosystem modellers) are less aware of these issues. Thus though some

of our results may not seem especially exciting to an ocean colour expert at first glance, we note these results could be of much interest in an interdisciplinary context.

Ocean colour satellite-derived Chl-a is often used as an evaluation tool for numerical models, and has been used for data assimilation (e.g. Gregg, 2008; Ciavatta et al. 2011, 2014; Rousseaux and Gregg, 2012). The likely biases in the Chl-a estimates are often not appreciated by the modelling community: Modellers sometimes mis-interpret mismatches that are actually potentially due to product biases, or worse have tuned their models or assimilated the products to capture the ocean colour derived Chl-a even where it is likely biased. There is also an inherent disconnect between model output and ocean colour products. Most biogeochemical models have a base currency of carbon, some have a dynamically varying phytoplankton Chl:C, very few resolve spectral irradiance, and even fewer resolve reflectance. However, there are some models that have recently incorporated more thorough treatment of the light field (e.g. Gregg and Casey, 2007; Mobley et al, 2009; Dutkiewicz et al, 2015), and some now include aspects such as reflectance or water leaving irradiances that more directly relate to ocean colour (Dutkiewicz et al 2015; Baird et al, 2015; Gregg and Rousseaux, 2017).

By resolving variables that are similar to ocean colour measurements (e.g. reflectance) models can be used to help explore uncertainties in ocean colour products and potentially even to aid in algorithm development. Mouw et al. (2012) used diagnosed optical parameters offline using output from a numerical model to provide ocean colour like products such as reflectance. That study isolated the effects of chlorophyll concentration, phytoplankton cell size and size-varying absorption on remotely sensed reflectance. However, it is only recently that models have directly included the treatment of ocean optics to allow for explicitly including diagnostics such as remotely sensed reflectance (e.g. Dutkiewicz et al., 2015; Baird et al., 2016). Here we use one of these models, a global three-dimensional biogeochemical, ecosystem, and radiative transfer numerical model (Dutkiewicz et al., 2015), that can act as a virtual laboratory to explore the connections between satellite derived products and the ecosystem variability that they are attempting to capture. The model resolves sufficient details of the marine ecosystem, water optical constituents as well as explicit upwelling irradiance.

We first briefly describe the numerical model (Section 2), before calculating a "satellite-like" derived Chl-a product from the model spectral reflectance output and explore the potential biases that arise between derived and "actual" model Chl-a (Section 3). Here we focus only on the biases due to the choice of algorithm, and not from other uncertainties that arise in the real world Chl-a products. (In this article "real-world" will be used to refer to the real ocean and the derived ocean colour products that are provide by space agencies. The "real world" is thus different to the numerical biogeochemical/ecosystem/optical model output and the products derived from it. Additionally, when we use the word "model" in this article, we refer to the numerical biogeochemical/ecosystem/optical model: In the ocean colour community "model" often refers to bio-optical relationships, we do not use "model" with this definition here.) Section 4 examines the temporal mismatches that occur in the derived product. We specifically explore how other optically important constituents, such as coloured dissolved organic matter (CDOM), detrital particles and accessory pigments limit the performance of the algorithm (Section 5).

This paper provides a proof of concept for using numerical model output to explore uncertainties and biases in information derived from surface ocean colour by specifically considering the potential uncertainties in the frequently used blue/green

reflectance ratio algorithm for determining Chl-a. Here, using the knowledge of model "actual" Chl-a, other optically important and reflectance at every location and every day allows us to examine these uncertainties and their causes more completely than is possible in the real world with its limited in situ observations.

## 2 The biogeochemical/ecosystem/optical model: Description and Results

We use a biogeochemical/ecosystem/optical numerical model as configured in Dutkiewicz et al (2015). We provide a brief description of the pertinent features here but refer the reader to that paper for more details, equations, parameter values and evaluation. The model resolves the cycling of carbon, phosphorus, nitrogen silica, iron, and oxygen through inorganic, living, dissolved and particulate organic phases (including CDOM). The biogeochemical and biological tracers are transported and mixed by the MIT general circulation model (MITgcm, Marshall *et al.*, 1997) constrained to be consistent with altimetric and hydrographic observations (the ECCO-GODAE state estimates, Wunsch and Heimbach, 2007). This three dimensional configuration has coarse resolution (1°×1° horizontally) and 23 levels ranging from 10m in the surface to 500m at depth. We resolve 9 phytoplankton functional types (diatoms, other large eukaryotes, coccolithophores, pico-eukaryotes, *Synechococcus*, high and low light *Prochlorococcus*, *Trichodesmium* and unicellular diazotrophs) and two grazers. These phytoplankton types differ in the types of nutrients they require (e.g. diatoms require silica), maximum growth rate, nutrient half saturation constants, sinking rates, and palatability to grazers. The phytoplankton also differ in their spectral absorption and scattering (see Figure 1 in Dutkiewicz et al., 2015) and maximum Chl-a:C. The different scattering and absorption spectra for each functional group incorporate the packaging effect (e.g. diatoms have a flatter absorption spectrum than the pico-phytoplankton), but we note that the model does not incorporate changes in the shape of the absorption or scattering spectra due to temporal photo-acclimation. The phytoplankton have dynamic Chl-a:C ratios that change with light availability, temperature and nutrient stress following Geider et al (1998). Thus the model explicitly resolves the Chl-a content of each of the 9 phytoplankton types as model state variables. The sum of this dynamic Chl-a across all phytoplankton types will be referred to as model "actual" Chl-a in the rest of this manuscript.

This model also explicitly includes radiative transfer of spectral irradiance in 25nm bands between 400 and 700nm. The three stream (downward direct, $E_d$, downward diffuse, $E_s$, upwelling, $E_u$) model follows Aas (1987), Ackelson et al (1994), and Gregg (2002), though here it is reduced to a tri-diagonal system that is solved explicitly (Dutkiewicz et al., 2015). The model captures the spectral absorption and scattering properties of water molecules, the 9 phytoplankton types, detritus and CDOM. It does not however include additionally potentially important components such as minerals and viruses (Stramski et al. 2001) or salt (Werdell et al., 2013b). Irradiance just below the surface of the ocean (direct, $E_{do}$, and diffuse, $E_{so}$, downward) is provided by the Ocean-Atmosphere Spectral Irradiance Model (OASIM, Gregg and Casey, 2009).

The model was run for 10 years for a recurrent "typical" year and then with interannual forcing from 1992 to 2006. Model output compares well to in situ and satellite-derived biogeochemical and ecosystem observations (Dutkiewicz et al., 2015). In

particular the magnitudes and patterns of absorption and scattering of different water constituents are captured along the Atlantic Meridional transect cruise (AMT15), as well as the spectral penetration of irradiance and key aspects of the community structure. Model "actual" Chl-a (the sum of the time varying Chl from each of the 9 phytoplankton types resolved) captures the regional patterns seen in a satellite-derived Chl-a. (Here we use the Ocean Colour Climate Change Initiative (OC-CCI)

project V2 product). As noted (and discussed more fully) in Dutkiewicz et al (2015) there are biases between the model and the observations, in particular larger values in the Southern Ocean and seasonally in the North Pacific than in the real-world satellite-derived Chl-a (Fig 1a,b,d,e).

The numerical model provides spectral surface upwelling irradiance: output that is similar to measurements made by ocean colour satellites. We calculate model subsurface reflectance for each waveband as the upwelling irradiance just below the

surface (all diffuse) divided by the total downward (direct and diffuse) irradiance also just below the surface:

$$R(\lambda, 0^-) = \frac{E_{uz}(\lambda)}{E_{doz}(\lambda) + E_{soz}(\lambda)},$$

where the $z$ in the subscript indicates that the irradiance has been re-computed using OASIM code for a zero solar zenith angle to compare more directly to observed normalized reflectance. Satellite sensor measurement (e.g. NASA and ESA products) have been normalized such that they are projected as if there was zero solar zenith angle.

To compare to satellite products, we first convert from irradiance reflectance to remote sensing reflectance using a bidirectional function Q:

$$R_{RS}(\lambda, 0^-) = R(\lambda, 0^-)/Q.$$

The bidirectional function Q has values between 3 and 5 sr (Morel et al., 2002) and depends on several variables, including inherent optical properties of the water, wavelength, and solar zenith angles (Morel et al., 2002; Voss et al., 2007). Here for

simplicity we assume that Q = 3 sr (see Appendix A for discussion of this assumption  and for evidence that the choice of Q makes little difference to model results). We note that Gregg and Rousseaux (2017) make a similar choice of a constant Q. Secondly we convert to above surface remotely sensed reflectance using the formula of Lee et al (2002):

$$R_{RS}(\lambda, 0^+) = 0.52 R_{RS}(\lambda, 0^-)/(1 - 1.7 R_{RS}(\lambda, 0^-)).$$

Hereafter we will refer to this quantity as $R_{RS}$.

We compare the model output to real world remotely sensed reflectance using the OC-CCI product (Fig 2). We note that the model does not have the exact same wavebands as any of the ocean colour satellites, and as such here we compare to the nearest bands: 450nm model to 443nm for the OC-CCI product, and 550nm model to the 555nm OC-CCI product. The model captures the reversed patterns between blue (443nm/450nm) and green (555nm/550nm) $R_{RS}$ between gyres and high productive regions. The model blue $R_{RS}$ (Fig 2a,b,c,d) captures the spatial and seasonal patterns in the real world satellite product.

However, the model has lower blue $R_{RS}$ in the southern Pacific gyre in January. We note though that the model lowest Chl-a in this region is offset from the real-world OC-CCI product (Fig 1a,b). Similarly the model blue $R_{RS}$ is too high in the equatorial Atlantic and Pacific, but where the model Chl-a is likely too low relative to the real world Chl-a product (see Fig 1). The model has noticeably higher green (550nm) $R_{RS}$ in the equatorial Atlantic and Indian than the satellite measurements but note that

these are regions of high cloud cover where the real world satellite product may be biased. We also find higher green $R_{RS}$ (Fig 2 e,f,g,h) in the North Pacific, but this might be due to model Chl-a being too high in this region (see Fig 1). In general the differences between model and the real world satellite $R_{RS}$ appear often to be linked to discrepancies between the model and real world satellite derived Chl-a product (and likely also in situ measurements). The model blue and green $R_{RS}$ appears to be consistent with the model actual Chl-a fields in a way that is similar to the real world and as such we believe appropriate and useful to use these model remotely sensed reflectance ("model ocean colour") to construct "satellite-like-derived" Chl-a using the blue to green reflectance ratio algorithm.

## 3 Constructing "satellite-like" derived Chl-a

We follow the blue/green reflectance ratio methods used to derive Chl-a from surface reflectance (e.g. O'Reilly et al., 2000). We first determine the log of the blue/green reflectance ratio: $X = \log(R_{RSB}/R_{RSG})$, where $R_{RSB}$ is the largest of the reflectance at 450m, 475nm, or 500nm at any location and $R_{RSG}$ is the reflectance at 550nm. We calculate $X$ using the daily model output from 1992 to 2006. We exclude any grid locations with daily mean PAR is less than 15 μEin/m$^2$/s (see Appendix A for explanation of this cutoff), with "actual" Chl-a less than 0.01 mg Chl/m$^3$ or with depths less than 1000m since the coarse resolution model does not adequately resolve coastal dynamics.

The blue/green reflectance ratio method uses a 4th order polynomial such that the derived Chl-a ($chl_d$) is:

$$chl_d = 10^{a_0 + a_1 X + a_2 X^2 + a_3 X^3 + a_4 X^4} \qquad\qquad \text{Eq 1}$$

The key here is to determine the best coefficients $a_0$ to $a_4$. We use a least squares fit to find $a_0$ to $a_4$ using three different approaches in our model "virtual laboratory".

### 3.1 Approach 1: Global Coefficients using Subsampled Fields (GS)

The first approach follows that used in real-world algorithm development (e.g. OC4 for SeaWiFS and OC-CCI, OC3M-547 for MODIS). The NASA bio-Optical Marine Algorithm Data set (NOMAD, Werdell and Bailey, 2005) was constructed from coincident radiometric observations and phytoplankton pigment data and has been extensively used for satellite derived Chl-a (and other) algorithm development. For direct comparison between real-world and emergent within-model relationships we therefore sub-sample the model "actual" Chl-a and reflectance ratio, X, at locations and dates nearest in time and space those in NOMAD. The resulting relationship between model blue/green reflectance ratio (X) and Chl-a from subsampling the model (Fig 3a) is similar to that found for real-world algorithms (Fig 4, Table 1). Some of the differences between real-world and model coefficients is likely to come from the use of different exact bands in the blue and green (e.g. 550nm for model green versus 555nm for OC-CCI). We note that this subsampling is highly biased to the low latitudes.

We use a least squares fit to find $a_0$ to $a_4$ from this subsampled dataset, the corresponding function is shown with a solid line (Fig 3a). We then used these coefficients and X from every grid cell of the model to produce a model "satellite-like" derived

Chl-a (Fig 1c, 5a) for the entire model output (daily from each grid cell, about 140 million data points). This derived Chl-a is analogous to the real-world satellite-derived Chl-a product (e.g. the OC-CCI product). Differences in coefficients relative to those for real-world algorithms (Table 1) are not large and the function looks very similar to those for the real world (Figure 4). We note that while the model world is an idealised system (and hence differences to real world are to be expected) one advantage is that there are no errors on the properties themselves (in contrast to measurement uncertainties on in situ Chl-a and satellite-derived reflectance in the real world) so the model allows for a more precise interrogation of the algorithm biases by themselves.

The root mean square error (RMSE) between the model derived Chl-a and the model actual Chl-a is 0.48 mg Chl/m$^3$ (0.16 for log transformed output) and has an r$^2$ of 0.60 in linear space (0.91 in log transformed data, see Table 2). There are substantial errors at higher Chl-a (Fig 5a), which translate to large biases in the high latitudes (Fig 6a,d). Larger errors at higher absolute concentrations are anticipated given the polynomial fitting was done in log space. The mean value of the absolute bias for all occasions and times where the derived product could be calculated was 22%, though we find that over 35% of the open ocean points (in space and time) had less than 10% absolute error (Fig 7a). We find that the monthly biases have regionally distinct patterns (Fig 6a,d).

Finally in this section, we ask: Which model Chl-a (derived versus actual) best matches real-world OC-CCI product? We did not do this for model validation purposes (see evaluation in Dutkiewicz et al., 2015), but rather to re-emphasis that the satellite derived Chl-a products are proxies for real world actual Chl-a: The two are not the same thing. We compare climatological monthly model derived Chl-a and model actual Chl-a to OC-CCI monthly climatology regridded to the model configuration (1 degree resolution). We find that the model derived Chl-a has global RMSE of 0.29mg/m$^3$, which is significantly lower than 0.64mg/m$^3$ found when comparing model actual Chl-a to OC-CCI. Comparisons are particularly better for the Southern Ocean and North Pacific (Fig 1). Consequently, some (though certainly not all) of the biases noted when comparing model actual Chl-a (Fig 1b,e) to real world satellite derived Chl-a products (Fig 1a,d, section 2 and in the model evaluation done in Dutkiewicz et al. 2015) are due to the real world Chl-a derived product bias and not a deficiency in the biogeochemical/ecosystem/optical model. It follows that a model satellite-like derived products (Fig 1 c,f) might be a better evaluation tool for comparing to ocean colour products derived with the same algorithm (Fig 1a,d) than the model actual Chl-a fields themselves.

### 3.2 Approach 2: Global coefficients using output from all locations (GA).

Secondly, we tested whether a lack of data to train the algorithm leads to some of the large errors in the derived Chl-a. We used model output for every surface grid cell and for each day (about 140 million points) to train the algorithm (Fig 3b). We note this is a purely hypothetical exercise: if one knew the Chl-a at every point and every day, why would one need to derive the Chl-a from a proxy (X)? However, here we are asking rather: Given almost perfect knowledge of Chl-a and X, what is the best that a global set of coefficients for the algorithm given in Eq 1 can do in capturing the actual Chl-a? In other words, even

given perfect training dataset (and in an idealised model virtual world), how good could the global OC4-style algorithm possibly be?

In contrast to sub-sampling the model (approach GS), when the full model output is included the relationship between X and actual Chl-a shows considerably more scatter and reveals a distinct cluster below the main body of points at low Chl-a (Fig 3b, the second "tail" below the main cloud). Although this cluster contains only a minority of points (less than 0.003% of the points), the mismatch is of interest and will be discussed in Section 5. Though the coefficients for the algorithm for the full dataset are different (Table 1), the fit is very similar at low Chl-a, but diverges at intermediate and high Chl-a (see solid and dashed line in Fig 3b). When comparing derived and actual Chl-a the GA coefficients lead to a better $r^2$ (Fig 5b, Table 2) than were achieved using the subsampled algorithm (GS). Though there is improvements in some regions in the higher latitudes, there is actually decrease in skill at lower latitudes (Fig 6b,e compared to a,d). There is in fact a slight increase in the mean % absolute bias (23%) between this and the GS estimates: When transformed into percent errors the increased biases at low Chl-a, low latitude regions become more prominent.

Not surprisingly our results suggest that a 4$^{th}$ order polynomial with one set of global coefficients will not in fact be able to fit both high and low concentrations accurately, no matter how much "data" is available to train the algorithm. Thus, though getting more in situ data in the ocean will still be beneficial for future algorithm development, the use of a single set of coefficients, derived from an improved in situ dataset, used over the whole globe is not likely to significantly improved biases everywhere.

**3.3. Approach 3. Regional Coefficients (RA)**

Recognizing that waters can have distinct optical properties (Moore et al., 2009; Szeto et al, 2011), there have been several projects to produce regionally distinct algorithms (e.g. Szeto et al., 2011, Johnson et al., 2013, latest release (V3) of the OC-CCI project, https://www.oceancolour.org). Here we take this concept to the extreme and construct a set of coefficients for each grid cell in the numerical model. We use the algorithm function as provided in Eq 1 and find the coefficients for each location using output from every day over 15 years. Here, we are testing whether X and Chl-a co-vary over time at each location, as opposed to over both time and space as in the previous two approaches (GS, GA). As with the global algorithm described in Section 3.1 and 3.2 we exclude any grid locations with daily mean PAR less than 15 µEin/m$^2$/s, with "model actual" Chl-a less than 0.01 mg Chl/m$^3$, and where depths are less than 1000m We also exclude any grid cells where the output falls outside these cut offs for more than half the year.

The regional specific algorithms provide a better Chl-a product with a significant reduction in the bias (Fig 6c,f), $r^2$, and RMSE (Fig 5c, Table 2). The mean absolute bias of all places and occasions where the derived product can be calculated is 17%, lower than the global approaches (GS, GA), and more than 50% of the model output points having less than 10% error (Fig 7c). Unsurprisingly, when averaged over the full time period, the regional algorithms at every location performs better than either of the global algorithms. However, there are still significant seasonal biases (Fig 6 c,f) (discussed more below). Note

that the biases switch sign between seasons, such that the annual mean bias is extremely low. There are some locations where at some times in the year there is even less accuracy with regional approaches, as seen by the cloud of points at low derived Chl-a (Fig 5c). This indicates that in some regions Chl-a and X do not vary coherently over time and/or that Chl-a and X do not vary in a similar way to the global relationship (noting that the global relationship incorporates both temporal and spatial variability). In these locations the actual Chl-a at some times in the year is closer to the global 4[th] order polynomial (Eq 1) than the local. We note also that there is still a cloud of points below the main cluster (where derived Chl-a is higher than model actual) as was found for approaches GS and GA (Fig 5).

## 4. Temporal Considerations

We have noted that in all approaches, though even more obvious in RA, there is a seasonally altering pattern between the derived and actual model Chl-a (Fig 6). The amplitude of the peak of spring blooms is often underestimated in the products derived using global coefficients (GS and GA) in high latitude, especially in the subsampled algorithm (GS) (Figs 6). Derived Chl-a values were also often higher than model actual Chl-a outside of bloom peaks. We consider the phenology, using a single location (in the subpolar North Atlantic) for a single year as illustration (Fig 8a). Though the derived products show similar (though smaller) peaks to the actual Chl-a, and sometimes similar peak timing early in the season (see for instance the first distinct peak in this illustrative location), there are noticeable lags for the maximum peak (shown with a vertical dotted line) and other mismatches later in the season. We also find that the bloom period lasts later into the year. The actual Chl-a also starts its sharp increase in spring (the initiation of the spring bloom, shown with dashed line) considerably before all three derived products (Fig 8a). We follow the approach of Cole et al (2012) for determining the "initiation of the spring bloom" as the time when the Chl-a first increases 5% above the annual median (horizontal dashed line, more description in Appendix A). Figure 8 shows just one location for 1 year. To consider the large scale patterns, we determine the lag in the spring initiation (Fig 9a) and maximum bloom timing (Fig 9b) for each location averaged over all years. We find that in almost all locations the derived Chl-a shows the bloom starting later than the model actual Chl-a (Fig 9a). This offset is typically by about 5-10 days but can be as much as 30 days. The maximum Chl-a from the derived product also lags the actual Chl-a in most locations, though by only a few days (Fig 9b). These results indicate that temporal as well as spatial biases occur as a result of deriving Chl-a from X and suggests care should be taken when calculating phenology from satellite products or when evaluating phenology in models using satellite-derived Chl-a. We discuss the reason for the lags in the next section.

## 5. The Role of other optically important constituents

Chl-a is not the only optically important constituent in seawater. Phytoplankton have a variety of accessory pigments that lead to large difference in their spectral absorption. Additionally different morphologies and structures lead to a variety of scattering

spectra (see e.g. Fig 1 in Dutkiewicz et al., 2015). CDOM and detrital particles also absorb more in the blue than the green. How do these other optically important constituents affect the ability of the blue/green ratio algorithm to accurately estimate "in situ" Chl-a? Studies have indeed suggested that second order variability in ocean colour -derived Chl-a can be tracked to the effect of CDOM and non-algal particles (e.g. Loisel et al, 2010; Brown et al., 2008; Siegel et al, 2005a; 2005b). Here, using the knowledge of all constituents in the default experiment (discussed above) in time and space, we can examine the importance of the optically important constituents on model reflectance more thoroughly than is possible in the real world (albeit in the simplified model ocean) and also perform a series of sensitivity experiments targeting individually the other optically important properties.

There is a close connection between CDOM, detrital matter and Chl-a (Fig 10). In general most model data points lie on a linear line: higher Chl-a is closely linked with higher CDOM and detrital matter. The co-variation between CDOM/detrital matter is however not perfect, as has been noted in the real ocean (see e.g. Bricaud et al., 1981; Kitidis et al., 2006; Morel et al., 2010; Siegel et al., 2005b). In the model output there is significant scatter around the core linear relationship (Fig 10). In particular high CDOM can be associated with a wide range of Chl-a concentrations. On the one hand the co-variability between Chl-a, CDOM and detrital matter might help the reflectance ratio algorithm since all absorb more in the blue than the green. However, we find that though linked, there are noticeable lags in the sharp increase in accumulation (Fig 8b, Fig 9 c,d) and peak timing and decline(Fig 8b) between CDOM and detrital matter and the model actual Chl-a. Since CDOM and detrital matter are a product of primary production, there is a lag in the high latitude spring between the accumulation and peak of CDOM, detrital matter and Chl-a. It is the lag in the accumulation of the different constituents that causes the algorithms to struggle to get the phenology accurate. Moreover detrital particles also lag in their removal, and CDOM (which has relative long remineralization timescales) remains relatively high throughout summer and fall. This leads to the later decrease in derived Chl-a than actual Chl-a seen in Fig 8a. The algorithms also all overestimate the background model actual Chl-a. This result may not seem surprising to those in the ocean colour community. In fact, the role of CDOM as an independent tracer has led to the suggestion that CDOM could be used to track mixing in the deep ocean (Nelson et al., 2010). That there is a difference in timing between peaks and CDOM and Chl-a is also known (see e.g. Fig 8 in Nelson et al., 2013). However to our knowledge this is the first time a numerical model has been used to pull apart the differences in timing of the different constituents and that impact on phenology from an algorithm derived Chl-a product.

We add the caveats that the exact definition of "initiation of bloom" does impact how much of a lag there is in the phenology. For instance, if the first peak in the model actual Chl-a in Figure 8a was defined as "the spring bloom" we would suggest the derived Chl-a does capture the timing better (though not the magnitude). We also note that the model parameterization of CDOM and detrital particle are not necessarily sufficiently well developed to make quantitative statements on the likely real-world lags. Thus, though we do suggest there could be significant lags in phenology in the real world satellite Chl-a product, we do not suggest that the values in Figure 9 are necessarily accurate for the real world. This analysis should instead be seen as a cautionary statement about using satellite-derived products for phenology of the quantities for which they are proxies.

That the other optically important constituents lead to a mismatch in derived and actual Chl-a leads us to ask the question: Would the algorithms work better if there was not a variation spatially or temporally in detrital matter, CDOM or accessory pigments (and phytoplankton community structure)? The accessory pigments and the absorption and scattering spectra differ between phytoplankton types and hence the community structure affects the reflectance ratio. However, how the community

structure (co-)varies in relation to total Chl-a, and the corresponding combined effects on reflectance, are complex.

To explore how these other constituents affect the algorithm, we perform three sensitivity experiments. Each experiment is performed similar to the "default" run (a 10 year spin-up, 1992-2006 interannually varying component) and we construct $4^{th}$ order polynomials equivalent to Eq 1 using the subsampling approach (GS) for each experiment and derive Chl-a in each case. However, given computational and storage constraints we used monthly averaged values of Chl-a and $R_{RS}$ to calculate the

algorithm coefficients in these experiments rather than daily values (see Appendix B for discussion). We compare the results from these experiments (Fig 11, Table 3) to the GS results from the default run (i.e. the Chl-a derived product using subset of the data to find the algorithm coefficients, i.e. most like the real satellite product) also using monthly values for consistency.

*a) EXP-1 - aCDOM*: This experiment was the same as the default, but $a_{CDOM}$ was (artificially) set to uniform constant values,

specific for each waveband (e.g. 0.016 m$^{-1}$ for 450nm, approximately a globally mean). The constant $a_{CDOM}$ leads to substantial differences in biogeochemistry and community structure (see Dutkiewicz et al., 2015). We construct the satellite-like Chl-a using the approach explained above (Fig. 11b, compare to 11a). The algorithm derived Chl-a compares better to the model actual Chl-a by some metrics (see Fig 11b), but not all, than they did in the original experiment ("default") (Fig 11a). Thus the correlation between Chl-a and CDOM can enhance the algorithm in some locations (see improvement at high Chl-a), but not

at others (e.g. at low Chl-a, see the cloud above the 1-to1 line). However, most noticeable is the lack of points below the main cluster at low Chl-a that was detailed in all other experiments and all types of algorithm approaches (Fig 5 a,b,c, and Fig 11a,c,d). The fact that this cluster of points does not occur in EXP-1 where there is no variability in $a_{CDOM}$, helps explain their origin. CDOM is photo-bleached in the surface waters but has long remineralization timescale at depth (Nelson and Siegel, 2013). As such CDOM concentrations (and hence $a_{CDOM}$) tend to be lower in the surface water and higher at mid-depths (see

e.g. Nelson and Siegel, 2013); a pattern that is captured in the model (see Fig 3 in Dutkiewicz et al., 2015). When Chl-a is low in the model during autumn, some deep mixing brings high un-bleached CDOM to the surface. This mixing in the highly seasonal regions is what leads to the cluster of points with lower blue/green reflectance ratio than is typical for the actual Chl-a concentration (in Fig 3b).

*b) EXP-2 -Detrital matter:* Similar to EXP-1 experiment, the absorption and scattering by detrital matter was set (artificially) to a constant mean value over the entire globe in this experiment. Detritus itself continued to vary, but the impact of detritus on the optics was as if it constantly had a concentration of 0.36 mmolC/m$^3$. Biogeography and community structure did change as a consequence of the difference in the irradiance. When we calculate the derived Chl-a in a similar manner as in the default with GS approach, we found the r$^2$ and RMSE are very similar to the default experiment (Fig 11c). However, as with EXP-1,

there are some times/places where the algorithm does better than in an ocean with varying optical signature of detritus (e.g. high Chl-a) and some where they were not. This suggest that detritus and Chl-a have a complimentary effect within the algorithm in many, but not all locations.

*c) EXP-3 - Differences in Phytoplankton Absorption and Scattering:* Finally, to explore the role of differing absorption and scattering properties of the different phytoplankton types, we conduct an experiment where all phytoplankton types were assumed to have the same optical properties: the mean of the different absorption and scattering spectra (see black lines in Fig 1 of Dutkiewicz et al., 2015), and same maximum Chl:C. Thus the phytoplankton are optically identical. The main biogeochemistry was similar between this simulation and default experiment, though there is some re-arrangement of the

phytoplankton communities as species specific absorption is important to their competiveness and biogeography (Hickman et al., 2010; Dutkiewicz et al, 2015). We find a substantially higher $r^2$ and lower RMSE for the derived Chl-a in this experiment. Thus the accessory pigments lead to a large scatter in the relationship between blue/green reflectance ratios and actual Chl-a, making the algorithm approach less accurate. Whether and how these known differences in absorption and scattering make it possible to differentiate species from optical measurements is a promising area of current research (e.g. IOCCG report 2014).

These experiments illustrate how variability in the different optical constituents in time and space lead to significant inaccuracies in deriving Chl-a from the blue/green reflectance ratio, yet on the other hand correlations also enhance the algorithm in many locations. Some of the results and, in particular the statistics are specific to the choices made for the fixed values of $a_{CDOM}$ and detrital concentration, as well as the mean spectra chosen for EXP-3. On the other, no matter the choice

of $a_{CDOM}$, EXP-1 is especially useful in elucidating the role of autumn mixing that can bring high CDOM to the surface and impacting the derived Chl-a signature. This is to our knowledge the first time such interactions and their impacts on satellite-derived products have been illustrated using a global biogeochemical model. We believe that similar experiments will be a useful tool in further studies, especially into exploring the impact of different phytoplankton absorption spectra.

## 6. Discussion and Summary

In this study we have used a global three-dimensional biogeochemical, ecosystem, and radiative transfer numerical model to explore how well the magnitudes and seasonal variability of Chl-a can be captured by a product derived from a reflectance ratio algorithm. The model outputs spectral surface upwelling irradiance that includes the effects of the scattering and absorption of optically important water constituents (phytoplankton, water molecules, CDOM and detrital matter), and as such we calculate a remotely sensed reflectance that compares to actual satellite data. We then construct a frequently used algorithm

to calculate a "satellite-like" Chl-a product from the model blue/green reflectance ratio. Given a complete knowledge of all these components and derived products in the model ocean, we can explore the uncertainties and cause more completely than is possible in the real world.

When the model algorithm coefficients are calculated from only a subset of data, similar to that which is available in the real world (e.g. NOMAD, Werdell and Bailey, 2005), the resulting function is similar to those used for SeaWiFs and MODIS Chl-a products (Table 1, Fig 4). Using this algorithm, the model derived Chl-a underestimates the actual Chl-a at high latitudes (Fig. 6a,d). Overall the algorithm has a mean absolute bias of 22% in capturing the actual Chl-a, but more than 35% of the

model output have less than a 10% absolute error (Fig 7a). However seasonally the errors can be substantially higher and the biases can shift from positive to negative (Fig 6a,d). This study only considers the errors involved in the algorithm development. It does not explore the other potential errors that arise in the real world, for instance from cloud cover, errors in atmospheric correction, instrument drift, and in situ measurement errors and does not resolve all the complexity in the real-world optical constituents. As such, the errors and biases we calculate are underestimates relative to the real world. However,

they do suggest that the error of 35% in ocean colour Chl-a estimates that had been desired (e.g. McClain et al 2006) is theoretically possible in many regions of the ocean.

We explored the potential to reduce the error by having a much larger dataset (i.e. knowledge of Chl-a at every location every day) for calculating the coefficients of the reflectance ratio algorithm. Although there was improvement in the bias at high Chl-a concentrations, there was also an increase in % bias at low Chl-a concentrations. It is perhaps not surprising that a single

set of coefficients for the whole globe will not produce an accurate Chl-a product, even with much improved coverage of data to train the algorithm.

Several studies have explored regional specific algorithms and have indeed found different sets of coefficients work better for different locations (e.g. Szeto et al. 2011, Johnson et al., 2013  Haentjens et al. 2017). To explore this using the model, we again assumed a large dataset (i.e. knowledge of Chl-a at every location and day) and calculated the coefficients for an

algorithm unique to each grid cell of the model. The improvement is large, reducing to 17% absolute bias, and almost 50% of the model output points have less than 10% error (Fig. 7c). Though this result is based on the "best case" scenario, it nonetheless suggests that significant improvements in detecting Chl-a from space will be possible with regional specific algorithms. However seasonal biases (Fig 6c,f) can be quite large (though they do cancel out over the course of the year, such that the annual bias is small). Thus temporal variations in Chl-a and reflectance ratio (X) are also not perfectly captured by a 4th order

polynomial.

Significantly, there is a mismatch between the timing of the spring bloom between any of the algorithm derived products and the actual Chl-a. In almost all seasonal regions the derived Chl-a products suggest the initiation (and peak) of the spring bloom occur later than the actual model Chl-a. We showed how this mismatch could be explained by the role of other optically important water constituents.

Because CDOM and detrital matter are also by-products of primary production and subsequent heterotrophic processes, they vary, at least in the surface ocean, in a similar manner to Chl-a (Fig 10) and have a similar effect on the blue/green reflectance ratio. The blue/green ratio algorithm has these co-variations intrinsically built into it (e.g. Morel 1988, 2009). Previous studies have noted that there are however discrepancies with this approach (Bricaud et al., 1998; Siegel et al, 2005a; Siegel et al., 2005b; Brown et al., 2008). In fact, the differences in how CDOM, and detrital matter absorb and scatter light has been used

in algorithm development (e.g. Sathyendranath et al., 1989; Roesler and Perry, 1995; Maritorena et al, 2002). The largest discrepancies between algorithms that explicitly include or exclude the differences in the optical properties is most noticeable at high latitudes (Siegel et al., 2005b), and CDOM and non-algal particles are noted to be especially important (Brown et al., 2008). Additionally it has been found that there are strong seasonal trends in variability of reflectance and reflectance ratios

(Brown et al., 2008). Here, we have used the model to show that indeed when the detrital matter and CDOM are distinctly decoupled from Chl-a there are stronger mismatches between actual and satellite-derived Chl-a. In particular we find that since CDOM and detrital matter in the surface water accumulate later in the spring than Chl-a (Fig 8b), the derived Chl-a increases and peaks later than the actual Chl-a (Fig 8a). Our model suggest that the timing of the spring bloom can be several days to weeks off when using satellite data to determine phenology (Fig 9).

CDOM can also muddy the signal at other times. CDOM is bleached in the surface waters (and is therefore mostly in low concentration) but is higher at depth where a long remineralization timescale allows it to accumulate. During the fall when in situ Chl-a is low, deep mixing can bring CDOM to the surface. At these times there is an anti-correlation between Chl-a and CDOM, and as such the algorithms tend to overestimate the Chl-a. This leads to the cloud of points where reflectance is lower than anticipated given the in situ Chl-a (Fig 3b), providing higher derived Chl-a than is actually there (Fig 5a,b,c). We suggest

that care should be taken when defining a fall bloom from satellite derived products given the effects of CDOM on reflectance during deep mixing.

It has been recognised that second order variability in reflectance spectra provides a potential method to determine phytoplankton species from space (e.g. Alvain et al, 2005). Using differences in the absorption and scattering spectra by various phytoplankton types to distinguish them optically is an important topic of research (e.g. IOCCG report 2014, see many

techniques cited therein; Werdell et al., 2014; Bracher et al., 2017). Here our model results echo this promising direction in showing that a large amount of the variability in the reflectance ratio versus Chl-a variability is due to the optical differences in phytoplankton (Fig 11d). In our sensitivity study, this appears in fact to have a larger effect than CDOM or detrital particles. The fact that temporal changes in the shape of the Chl-a specific light absorption and scattering spectra for each phytoplankton type does not vary with photo-acclimation in the model formulation means this result is likely under-estimated, though such

within-type variability is likely to have a small effect on sea surface reflectance compared to differences in spectra between types.

Chl-a derived from the model reflectance compares better to the OC-CCI Chl-a than the model actual Chl-a (Fig 1), with a significantly lower RMSE. These differences can particularly be seen in the high latitudes. This finding serves to highlight that Chl-a and reflectance-derived Chl-a are not the same thing and suggests that modellers should be careful in attempting to

30 compare too strongly with satellite derived Chl-a, especially in high latitudes where mismatches between derived and actual Chl-a are shown to be important. Biases are particularly noticeable in the Southern Ocean, which appears to have a very different optical signature (e.g. Szeto et al. 2011; Johnson et al., 2013). Our model results suggest that deep mixer layers (bringing high CDOM to the surface), a potentially different species composition (e.g. Ward, 2015), and high seasonality

(leading to mismatches in timing of the peaks in the different optically important constituents) leads these waters to have very different and seasonally varying optical characteristics.

The results presented here provide a novel assessment of the interactions between optical constituents, their effects on reflectance and derived Chl-a that compliment those that are possible for the real world (e.g. Moore et al. 2009; 2014; OCCI uncertainty products, http://www.esa-oceancolour-cci.org/?q=webfm_send/321). In the model all properties are known precisely everywhere allowing details of how the optical constituents and their interactions impacts reflectance and the derived products. However, the results should be taken in context of a modelling approach. The uncertainty estimates reflect only the subset of constituents and mechanisms resolved in the model and the model does not perfectly capture the Chl-a, optics or the reflectance algorithm. Thus the biases presented here should be considered qualitatively, rather than expecting the exact values and statistic to apply to the real ocean. The mismatch in bloom timing should also be interpreted in this way. It is unlikely that the model captures the correct lag between accumulation of CDOM and detrital matter. Thus the exact number of days that the derived Chl-a product lags the actual Chl-a is likely to be different in the real ocean. However the model captures enough of the real world to give insight into the interpretation of the ocean colour product.

A key motivation for this study was to demonstrate that a biogeochemical/ecosystem/optical model with radiative transfer component can be used as a laboratory to explore aspects of ocean colour. As such this study bridges between disciplines: particularly ocean colour and biogeochemical/ecosystem modelling. We believe that our approach could help modellers understand some of the limitations of ocean colour, something that is often lacking when their expertise in not in satellite measurements. We also hope that the ocean colour community will see the potential of model approaches such as this for deriving sampling strategies, further studies on newer Chl-a algorithms (e.g. NASA Reprocessing 2014.0, and OC-CCI V3 release), other ocean colour products, and will help with algorithm developments for current and future ocean colour measurements.

## Appendix A. Assumptions and Definitions

***Value of bi-directional factor, Q:*** The bidirectional function Q has values between 3 and 5 sr (Morel et al., 2002) and depends on several variables, including inherent optical properties of the water, wavelength, and solar zenith angles (Morel et al., 2002; Voss et al., 2007). We calculated model reflectance both with a constant value (3 sr) and with time/space/wavelength varying values calculated from the table of Morel et al. (2002). The differences in the relationships between Chl-a and blue/green reflectance ratio with variable and uniform Q was almost imperceptible (Fig 4). We used the constant/uniform Q (similar to that used in Gregg and Rousseaux, 2017) in this paper. However we note that the resulting values would only be slightly different if we had used the variable Q, and, in particular, the choice of Q would not have changed the interpretation and implications of our results.

***PAR cutoff:*** Satellite measurements of ocean colour cannot be obtained when irradiance fields are too low. These occasions occur during the winter in high latitudes. To compare better to satellite measurements, we choose to not include model data in similar conditions. We examined where and when satellite $R_{RS}$ (from OC-CCI) lacked data (due to low light) at the high latitudes, and found that the geographic locations and times matched well to when the OASIM input daily mean irradiance fields were less than 15 μEin/m$^2$/s (see Fig 1,2) . We thus used this value as a cutoff for calculating derived Chl-a.

***Determining the Initiation of the Spring Bloom:*** We found that the determining the spring bloom peak was quite noisy such that it was more informative to consider "initiation of the spring bloom". We therefore compared the timing of the bloom initiation between the actual and derived Chl-a. Following the approach of Cole et al (2012), we first reset each "year" at each grid cell by centring to the peak model actual Chl. We then determine when the model actual Chl-a reaches 5% above the annual median value. We define this as the actual "initiation of the spring bloom". To determine the lag in the initiation (Fig. 9) we calculated the day that the GS derived Chl-a product (that is closest to the real-world satellite-like derived Chl-a. e.g. OC4-like) reaches 5% of its respective median values.

**Appendix B. Exploring impact of using monthly means to determine algorithm coefficients**

The daily values for 15 years at each grid point creates a very large datafile. Diagnostics with, and storage of, this large dataset becomes extremely computationally expensive. In order to conduct sensitivity studies we found that we needed to reduce this data set. Here we explore only outputting monthly means of model $R_{RS}$ and Chl-a and thus reducing the dataset by 1/30$^{th}$. We determined the algorithm coefficients ($a_0$ to $a_4$ in Eq 1) using monthly rather than daily means and subsampling for the GS approach. The resulting function (Fig 4, solid black) is similar at low and intermediate Chl-a, but does deviate at high Chl-a from the algorithm found using daily mean values (light blue line). The $r^2$ from this algorithm with coefficients defined with monthly means was also not quite as good as that found using daily means (see Table 2 and 3). However we found that the results were similar enough that we could obtain qualitative comparison between sensitivity experiments EXP-1, EXP-2, EXP-3 discussed in Section 5. We also note that the resulting two dimensional histogram (Fig 11) has far lower density when using 4 million relative to 140 million points. Though not perfect, using monthly output does allow us to perform EXP-1 through EXP-3 and still feel confident that the between experiment differences are robust.

**Acknowledgements**

We thank Michelle Gierach and Colleen Mouw for discussions early on in this work, and for their belief that we could "model ocean colour". The paper was significantly improved by comments from two anonymous reviewers and from the Associate Editor, Emmanuel Boss. We are grateful to Watson Gregg and Cecile Rousseaux for providing the input fields, output, and

code for the Ocean-Atmosphere Spectral Irradiance Model (OASIM). This work was funded by NASA-NNX13AC34G and NASA-NNX16AR47G. We acknowledge the Ocean Colour Climate Change Initiative (OC-CCI) for satellite products.

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

|  | $a_0$ | $a_1$ | $a_2$ | $a_3$ | $a_4$ |
|---|---|---|---|---|---|
| model (GS, subsampled) | 0.4507 | -2.6040 | -1.2876 | 6.5324 | -5.1420 |
| model (GA, all output) | 0.6588 | -3.2742 | 0.5860 | 3.2253 | -3.0903 |
| OC4 (SeaWiFS) | 0.3272 | -2.9940 | 2.7218 | -1.2259 | -0.5683 |
| OC3M-547 (MODIS) | 0.2424 | -2.7423 | 1.8017 | 0.0015 | -1.2280 |

**Table 1: Coefficients for model global derived algorithms (Eq 1) and for the SeaWiFS and MODIS default algorithms**

|  | Approach 1: GS | Approach 2: GA | Approach 3: RA |
|---|---|---|---|
| $r^2$ (log space) | 0.91 | 0.92 | 0.95 |
| RMSE (log space) | 0.16 | 0.15 | 0.12 |
| $r^2$ (linear space) | 0.60 | 0.77 | 0.83 |
| RMSE (linear space) | 0.48 | 0.37 | 0.31 |
| absolute % bias | 22% | 23% | 17% |

**Table 2: Results of comparison between model "actual" and model "satellite-like" derived Chl-a for the three algorithm approaches discussed in Section 3. Statistics are calculated for each grid and each day over 15 years, except for grid cells and times with low light, very low Chl-a and shallow regions (see text).**

|  | Default | EXP-1: uniform $a_{CDOM}$ | EXP-2: uniform $a_{det}$ | EXP-3: phytoplankton optical same |
|---|---|---|---|---|
| $r^2$ (log space) | 0.90 | 0.87 | 0.89 | 0.95 |
| RMSE (log space) | 0.17 | 0.16 | 0.17 | 0.12 |
| $r^2$ (linear space) | 0.54 | 0.63 | 0.60 | 0.75 |
| RMSE (linear space) | 0.44 | 0.38 | 0.40 | 0.26 |

| | | | | |
|---|---|---|---|---|
| absolute % bias | 21% | 20% | 23% | 18% |

**Table 3: Results of comparison between model "actual" and model "satellite-like" derived Chl-a for the sensitivity experiments discussed in Section 5. All "satellite-like" derived Chl-a was calculated using the GS approach. "Default" is the full experiment discussed in Section 3, but with monthly R$_{RS}$ used to calculate the algorithm coefficients. Statistics are calculated for each grid cell and each month over 15 years, except for grid cells and times with low light, very low Chl-a and shallow regions (see text).**

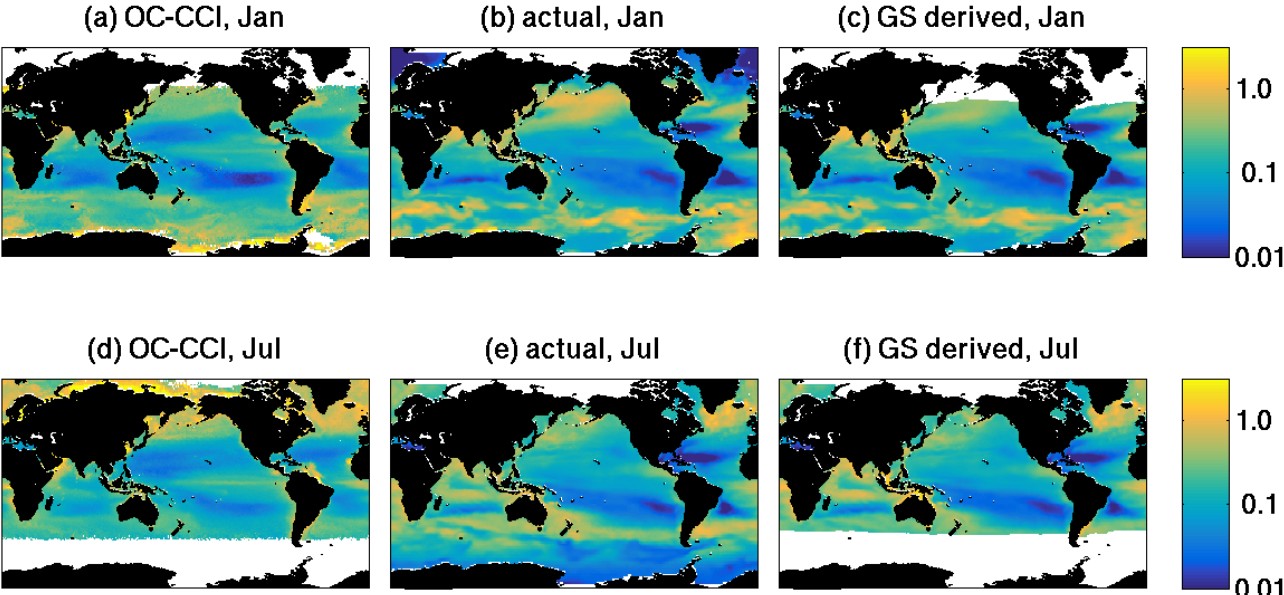

**Figure 1: Annual mean Chl-a (mg/m$^3$). (a) and (d) OC-CCI-derived; (b) and (e) default model "actual" 0-50m (summed over the 9 phytoplankton types); (c) and (f) default model "derived" (calculated from reflectance ratio and satellite-like algorithm trained with subsampled dataset, GS). Top row are January mean, bottom row are July mean. OC-CCI products (a and d) have no data when irradiances are too low. The model does not resolve the Arctic and thus there is not output here in (b), (c), (e), and (f). Additional lack of output in (c) and (f) indicates regions where PAR is less than 15 μEin/m$^2$/s. OC-CCI products were downloaded from https://www.oceancolour.org. We use version 2 of the OC-CCI, which uses an OC4 algorithm for determining the Chl-a product, and thus comparable algorithm as used in our model derived Chl-a shown in e,f.**

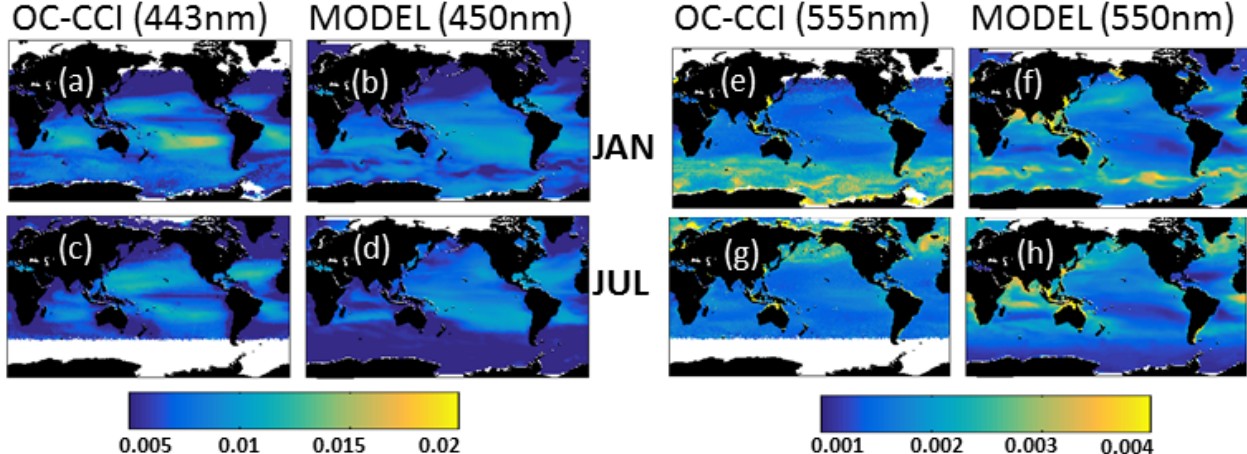

**Figure 2:** Remotely sensed reflectance (1/Sr) for (a) OC-CCI at 443nm, January; (b) model at 450nm, January; (c) OC-CCI at 443nm, July; (d) model at 450nm, July; (e) OC-CCI at 555nm, January; (f) model at 550nm, January; (g) OC-CCI at 555nm, July; (h) model at 550nm, July. ). We compare the model wavebands against the nearest OC-CCI wavebands, but note that they are not identical. OC-CCI products (a,c,e,f) have no data when irradiances are too low. For model lack of output indicates regions where PAR is less than 15 µEin/m²/s or the unresolved Arctic region. OC-CCI products were downloaded from https://www.oceancolour.org.

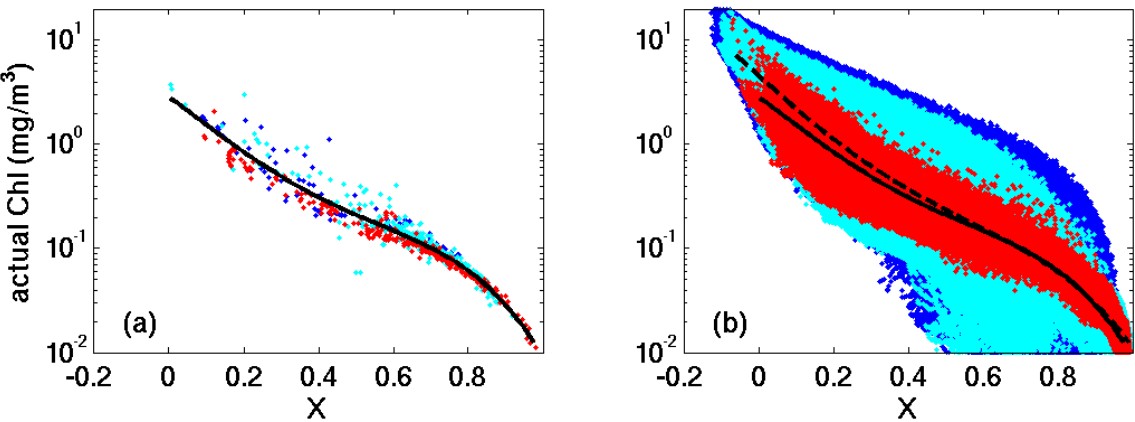

**Figure 3:** Model "actual" Chl-a and model blue/green reflectance ratio (X) for (a) subset of model output similar to that available from real world in situ observations (e.g NOMAD, Werdell and Bailey, 2005); (b) full model output (every day for 15 years from each grid cell, about 140 million points). Black solid line indicates the algorithm for $chl_d$ for where coefficients were determined from the subsampled datasets (GS), and in (b) dashed line is the algorithm where coefficients were calculated using the full dataset (GA). Dots are coloured red for locations equatorward of 30º, light blue for 30º to 60º, and dark blue for poleward of 60º.

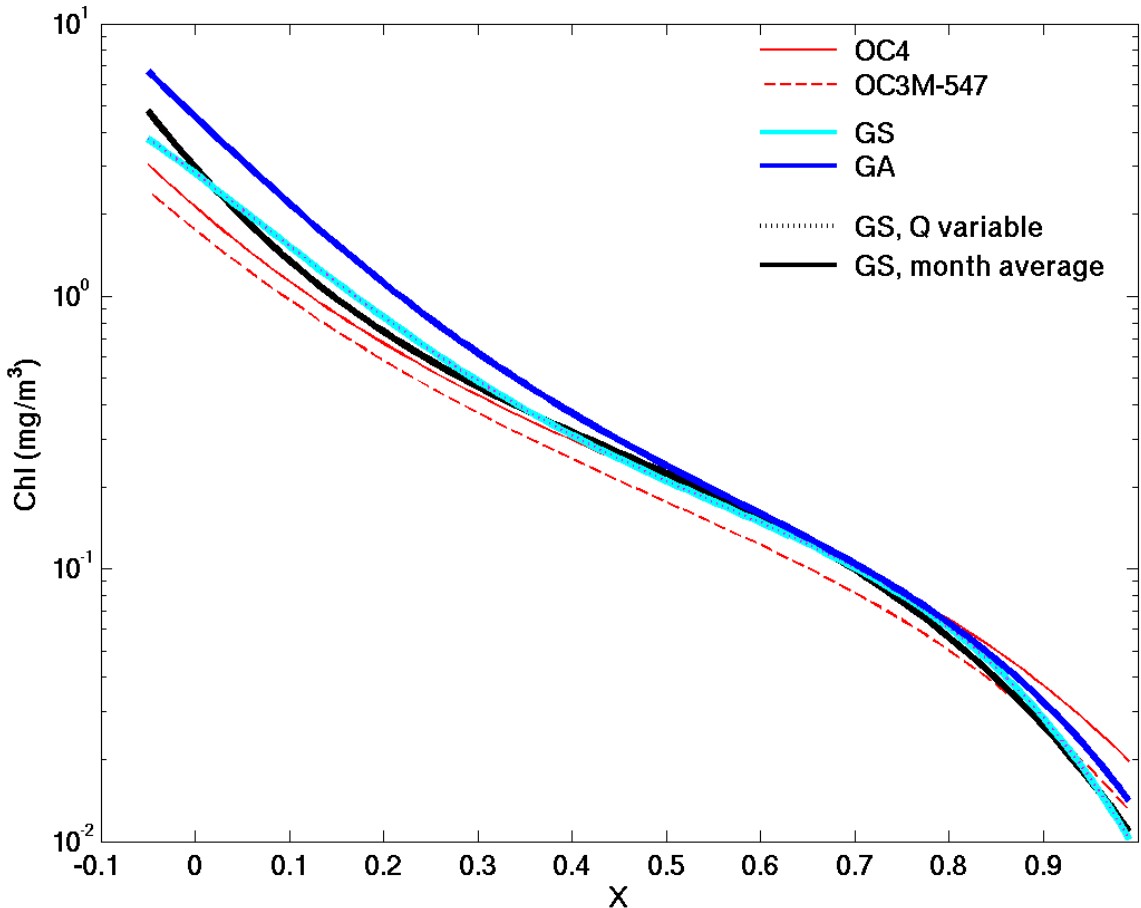

**Figure 4: Polynomials for Chl-a algorithm using the blue/green reflectance ratio. Shown are two real world algorithms: NASA OC4 (red solid, used in SeaWiFS and OC-CCI products) and NASA OC3M-547 (red dashed, used for MODIS product). Model algorithms shown are GS (light blue, same as in Fig 3a), where coefficients are found from subset of model output as dictated from real world in situ observations and GA (dark blue, same as dashed line in Fig 3b)where coefficients are found using the full data set. Also shown are two additional polynomials discussed in the Appendix: one found using a variable bi-directional coefficient, Q, and a subsampling of output as in GS (dotted black line almost exactly on top of the light blue line), and another where coefficients were found from a subsampling as in GS but using monthly average reflectance and Chl-a (black line). Note that the algorithms for the model come from band ratio of 425nm/450nm/475nm and 550nm. For the real world algorithms the band ratios are different and specific for the satellite sensor (SeaWifs or MODIS).**

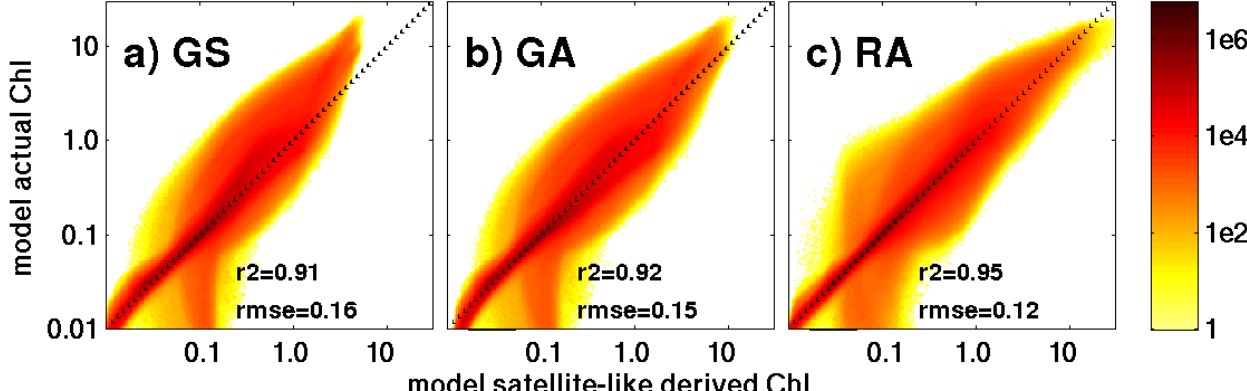

**Figure 5:** Two dimensional density histogram of model "actual" and model "derived" Chl-a using algorithm coefficients found for: (a) default experiment using approach 1 (global, subsampled output, GS), (b) default experiment using approach 2 (global, all output, GA), (c) default experiment using approach 3 (regional, RA), Dashed line indicates 1 to 1. Colour indicate the log of the fraction of all data that occur in phase space (the lightest yellow reflects a single instance in that bin). Statistics noted on the plot are for the log transformed output. The $r^2$ and RMSE in linear space is provided in Table 2.

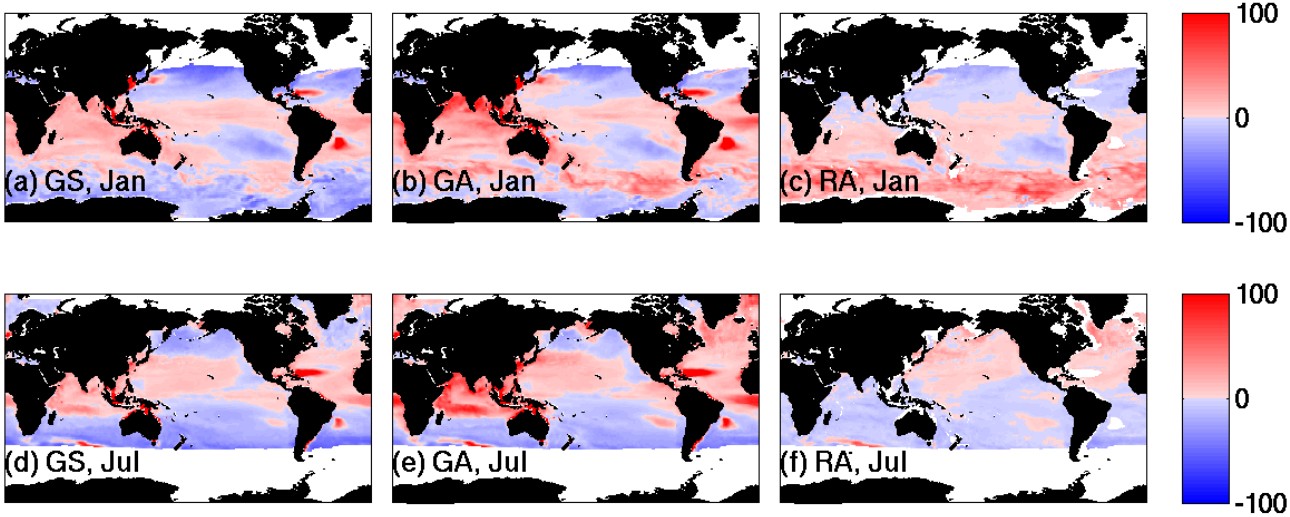

**Figure 6:** Percentage bias between monthly mean model "actual" Chl-a and model "derived" Chl-a ($chl_d$) using algorithm coefficients found for: (a,d) subset of output (GS); (b,e) full model output (GA); and (c,f) each grid cell (regional specific, RA). Top row is for January, Bottom row is for July. ). White areas indicate unresolved Arctic and regions where PAR is less than 15 μEin/m$^2$/s.

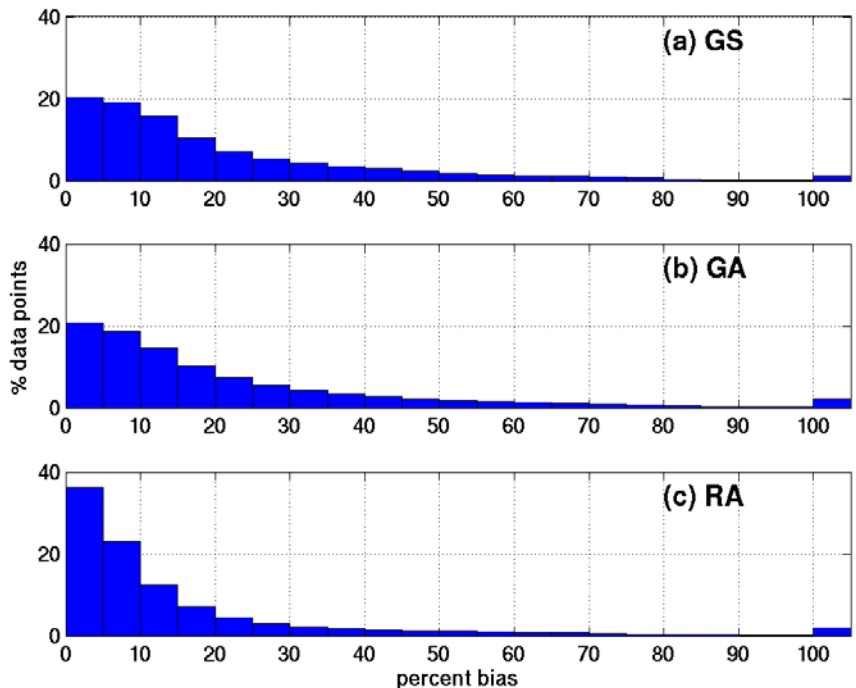

**Figure 7**: Distribution of percentage of model output (time and space, about 140 million "data" points) with absolute percent error between model derived Chl-a and model actual Chl-a for (a) global subsampled approach (GS), (b) global all output approach (GA), and (c) regional approach (RA).

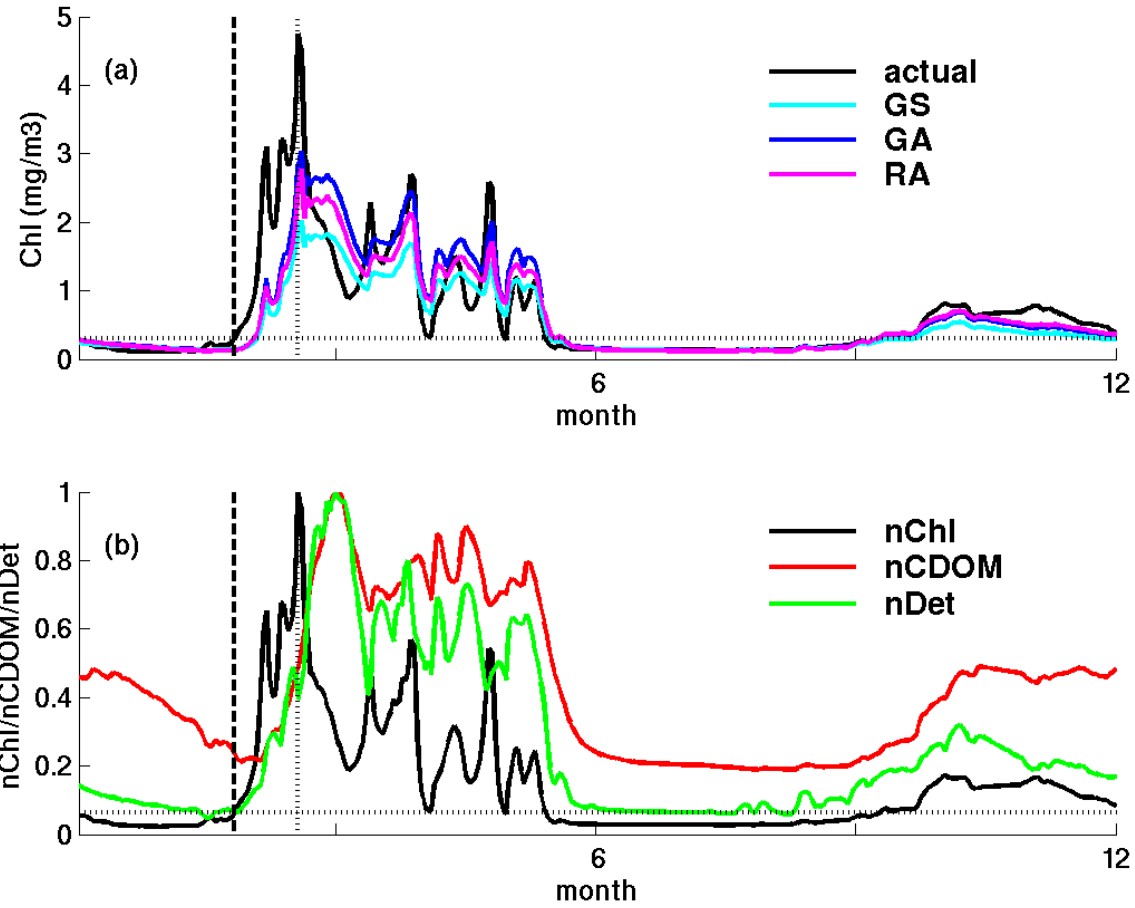

**Figure 8:** Illustrative example timeseries for one year from a single location in the North Atlantic (shown as x on Fig 9). (a) "actual" Chl-a (black), derived Chl-a using subsampled output (GS, light blue), derived Chl-a using all output (GA, dark blue), and the Chl-a product derived using a regional specific algorithm (RA, purple). (b) actual Chl-a (black), CDOM (red) and detritus (green), all normalized to their peak value. Dashed vertical line indicates the "initiation of the bloom" which is taken to be when Chl-a reaches 5% above the annual median value following Cole et al (2012) and discussed further in Appendix A  (dotted horizontal line shows this value for the model actual Chl-a). The vertical dotted line indicates the peak of the bloom. Shown here is only a single year and location, however for larger scale perspective, the difference in initiation and peak timing between model actual and derived Chl-a averaged over all years are shown for the globe in Figure 9.

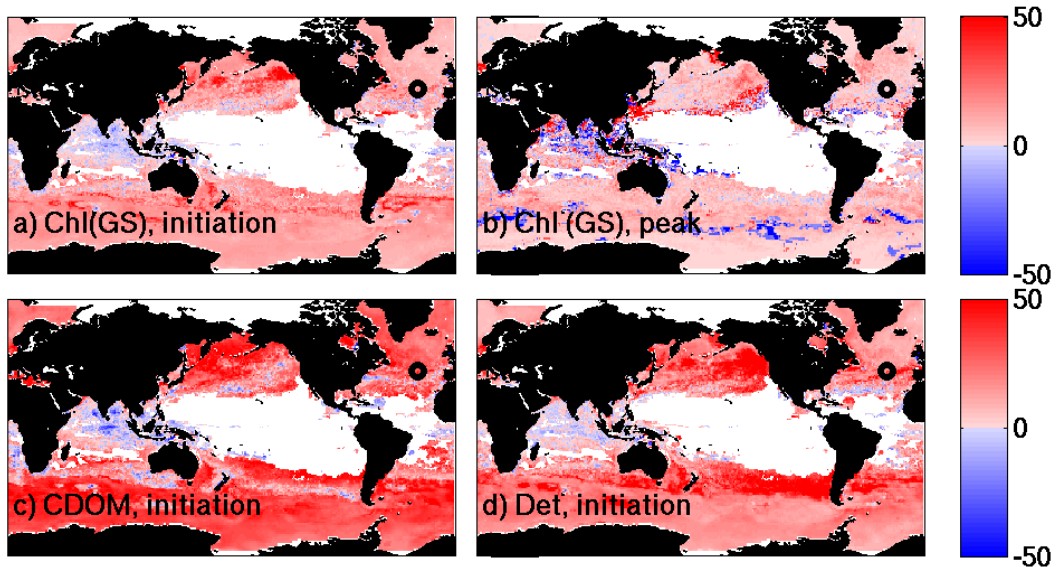

**Figure 9: Lag in phenology. Number of days between a) the initiation of the spring bloom from model actual Chl-a and that for the model derived Chl-a (GS);b) yearly maximum of model actual Chl-a and that for the derived Chl-a (GS); c) initiation of the spring bloom from model actual Chl-a and the initiation of the CDOM increase; d) initiation of the spring bloom from model actual Chl-a and the initiation of detrital particle increase. Bloom initiation is defined as when Chl-a, CDOM or detrital particles reach 5% above their annual median value (see Appendix A). White areas indicate regions with no significant seasonal cycle or are not resolved by the model (e.g. Arctic Ocean).**

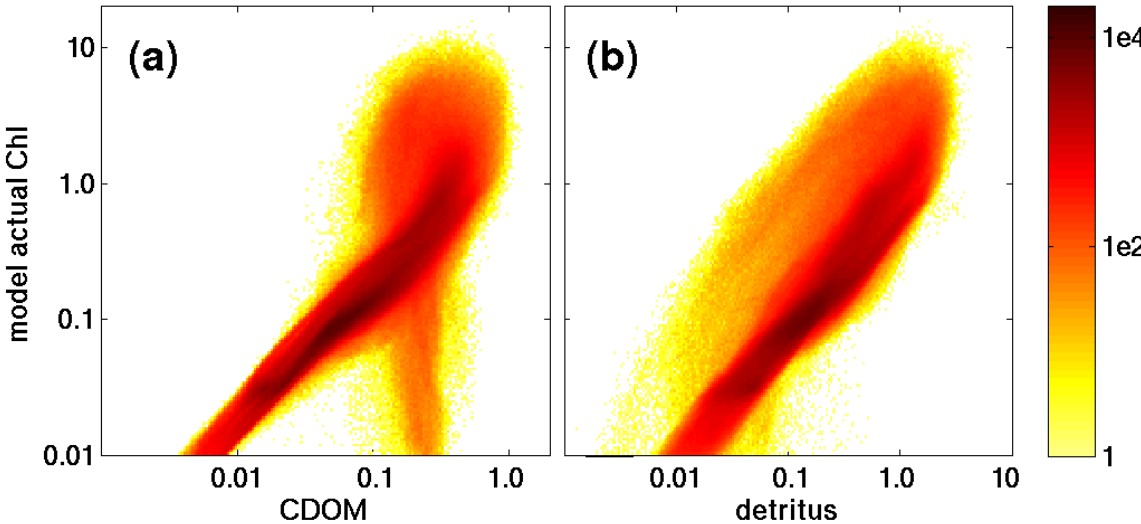

**Figure 10:** Two dimensional histogram of model output for "actual" Chl-a (mg/m$^3$) plotted against: (a) model CDOM (mmol C/m$^3$), (b) detritus (mmol C/m$^3$). Colour indicate the log of the fraction of all data that occur in bins in the phase space (the light yellow reflects a single instance in that bin).

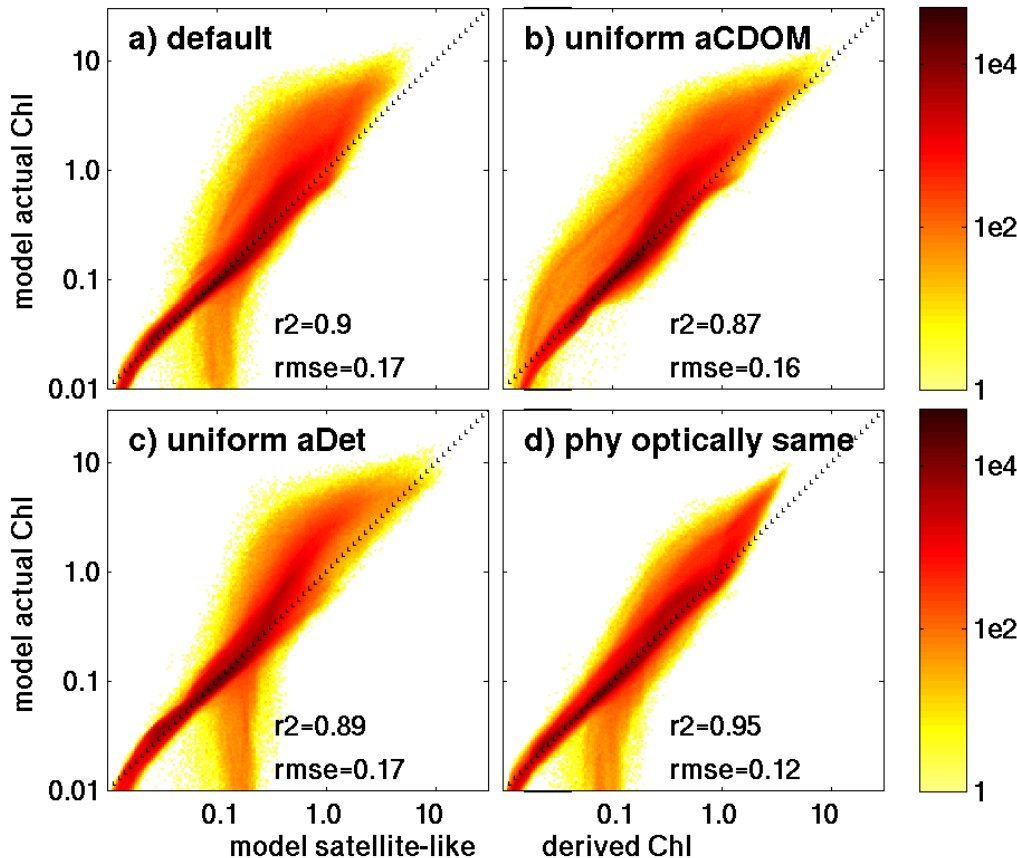

**Figure 11:** **Sensitivity Experiments. Two dimensional density histogram of model "actual" and model "derived" Chl-a using algorithm coefficients found for: (a) default experiment using approach 1 (global, subsampled output, GS), (b) EXP-1 (uniform and constant a$_{CDOM}$) approach 1, and (c) EXP-2 (uniform and constant a$_{det}$ and b$_{det}$) approach 1, (d) EXP-3 (no optical differences between phytoplankton) approach 1. Dashed line indicates 1 to 1. Colour indicate the log of the fraction of all data that occur in phase space (the lightest yellow reflects a single instance in that bin). Statistics noted on the plot are for the log transformed output. In. The r$^2$ and RMSE in linear space is provided in Table 3. In these plots, monthly mean output of Chl-a and R$_{RS}$ were used to calculate the algorithm, and only monthly mean output is shown (4 million versus 140 million points), thus at a great computational savings. The difference in the algorithm is shown in Figure 4 (the light blue line is the algorithm with coefficients found using daily values, versus the solid black line where coefficients where found using monthly values). Differences between 11a and 5a are due to this difference in sampling (discussed in Appendix B). Also notice the difference in values on the colourbars between this figure and Figure 5.**