# Peer review of "Modelling Ocean Colour Derived Chlorophyll-a"

_Biogeosciences, 2017_

## Referee Comment (RC1) · Anonymous Referee #1 · 28 Sep 2017

This manuscript provides an overview on the how a coupled biogeochemical-ecosystem-optical model can be used to explore ocean colour algorithms, with a focus on Chlorophyll-a. The authors effectively show the kind of interrogation studies that can be done with this type of "virtual laboratory". They clearly demonstrate how the ocean colour community can explore the bias and uncertainties of algorithms and their products, by investigating the effect of (1) other optically significant materials on derived Chlorophyll-a, and (2) different sized and regionally focused training datasets on robustness of an algorithm. I think this manuscript paves the ground for more detailed studies on the use of a radiative transfer component in a biogeochemical-ecosystem model to investigate ocean colour algorithms. The manuscript is well-written and logically presented, but there are a couple of points where I think a bit more clarity would

improve the presentation of the methods & results (see comments).

Specific comments:

P2 L20-21: the band-ratio definitely used to be the most commonly used Chl-a algorithm for NASA, but they switched their "default" Chl-a to a merged approach of Hu et al. (2012) and the OCx type algorithms in Reprocessing 2014.0. I am not suggesting you redo your analysis using the band-difference algorithm (because as I understand it, the point in the paper is more to show the kind of analysis you can do with this type of "virtual laboratory", and dealing with multiple Chl-a algorithms might confuse matters - that being said, it would be an interesting task), but I think it might be worth acknowledging that the OCx algorithms are not the most common for NASA anymore.

Hu et al. (2012), J Geophys. Res., 117(C1). doi: 10.1029/2011jc007395

P4 L29-30: While this appears to be true for the January images, it seems to me that the July OC-CCI image (1d) has higher values in the northern high latitudes (around Greenland, Bering Sea, around Scandinavia) than actual July image (1e).

P5 L15-27: It is a bit unclear to me which results we are comparing at different points in this paragraph e.g. are the "observations" (L19) the OC-CCI observations? What is the "real world actual Chl-a" (L24)? L19-20: Are you saying the model blue Rrs is too high in the equatorial regions compared to the OC-CCI, coincident with where the model "actual" Chl-a is too low compared to the OC-CCI? Are you meaning OC-CCI is the "real ocean"? Maybe this sentence could be reworded to clarify this.

P7 L4: I think this sentence could be more clearly explained. I think I understand the point you are making: that because the model derived Chl-a compares better with the OC-CCI Chl-a than the model actual Chl-a does, then some of the difference between OC-CCI and model actual Chl-a can be attributed to problems with the band-ratio algorithm (i.e. "product bias")? Is this what you mean by "product bias" - that there is an intrinsic problem with the band ratio formulation? I think the use of term "model" at the

end of this sentence is particularly confusing: often in the ocean colour community, the term "model" is used in terms of a bio-optical proxy/relationship e.g. Chl-a is modelled using the band-ratio. Perhaps use "ecosystem model" (or something similar), to make this distinction clear.

P10 L2-5 (& Appendix B): The exact method is a bit unclear to me here. Did you: take the results of the full run (i.e. those shown in Fig 5a), then take the monthly mean of Rrs output and Chl-a input and derive the 4th order polynomial coefficients on those monthly means? Then, for each of the 3 experiments, did you: do a full run with daily values, take the monthly mean of the model output Rrs and input Chl-a, and use the monthly band-ratio relationship with the monthly Rrs for input? Or did you set up the model with monthly means for the input? Perhaps this could be clarified in the text.

Fig 11: If I am understanding correctly, Fig 11a is the same as Fig 5a, but 11a uses the monthly coefficients i.e. the black line in Fig 4, whereas 5a uses the light blue line. I think it would be useful to point this out explicitly.

P12 L20-21 & L33-P13 L1: I'm not so sure it is quite as simple as this. I agree that the other optically significant materials are contributing to false Chl-a signals: there is a shoulder in all the derived Chl-a time series (Fig 8a), that aligns perfectly with the peak in CDOM and detritus (Fig 8b). But you can see the pattern of the actual Chl-a signal in the derived values, with peaks aligning on around days 60 and 75 - the magnitude of these derived values are just less than the actual values, but I'd say these are the "true peaks" of the spring bloom. After approximately day 75, there is the interference from CDOM and detritus, hence when calculating the initiation of the spring bloom (as described in the appendix), this large "false peak/shoulder" increases the median Chl-a and skews the determined initiation date. So I think what your data could be showing is (1) the Chl-a products do capture the peak of the spring bloom, but the magnitude of that peak is too small, and (2) the CDOM and detritus contribute to a false Chl-a signal, which makes it appear as if the bloom lasts longer and (depending on how you define bloom initiation) makes the initiation date appear to lag compared to the model actual.

It might be useful to have a table presenting the numerical results (e.g. log/linear RMSE, absolute % bias, etc.) for each approach in Section 3 and 5, to make it easier to compare the different results.

Fig 4 and Fig 8: Legends would make these plots easier to read.

Fig 5 and Fig 11: it would be useful to have a title or text on each graph to show which subplot is which e.g. "(a) GS"

Fig 8: The thick black line on top of the other time series signals masks some of the detail of the derived Chl-a products, particularly after the first 3 months - could this be represented in a different way? Also, check the axis labels, I think Fig 8b is showing days, not months.

Technical Corrections:

P1 L15-16: missing the word "to" i.e. sentence should read "...derived Chl-a to the actual..."

P1 L25: should be either "These results indicate" or "This result indicates"

P9 L29: Should this sentence not end with a question mark? i.e. "...community structure)?"

P11 L23: remove the second "like"

P12 L24: build should be built

---

## Referee Comment (RC2) · Anonymous Referee #2 · 29 Sep 2017

The paper 'Modelling Ocean Colour Derived Chlorophyll-a' by Dutkiewicz et al. uses a modified version of the MIT general circulation model that incorporates scattering and absorption properties of water, detritus, coloured dissolved organic matter (CDOM) and 9 phytoplankton types. 'Actual' chlorophyll-a concentration (Chl-a) is determined by summing the variable chlorophyll-a concentration of the 9 phytoplankton types. Remote sensing spectral reflectance, determined from the resulting modelled upwelling and downwelling irradiances are used in a NASA OCx type algorithm to compute satellite-like 'derived' chlorophyll-a concentration which is compared to 'actual' chlorophyll-a concentration. Firstly, the model output is used to test assumptions used in the derivation of chlorophyll-a concentration using the standard OCx ratio algorithm. Secondly, bloom initiation timings, determined from the 'derived' and 'actual' chlorophyll concen-

trations, are compared. Lastly, the impact that other optically important parameters may have on 'derived' chlorophyll concentration is explored. The authors conclude that (a) applying a single set of coefficients in the OCx algorithm globally is not as accurate as applying regionally variable coefficients, (b) there is a temporal mismatch between the initiation of the spring bloom defined using 'actual' Chl-a compared with that defined using 'derived' Chl-a and (c) that this mismatch may be caused by the optical influence of other substances such as CDOM and/or detritus.

I found this paper to be generally well-thought out and well-written. For colleagues who are not regular users of these products, this paper provides important caveats for the use of satellite derived data. For those of us who work with ocean colour derived products more often, the results may not be unsurprising, but it serves as a timely reminder of the limitations associated with satellite derived data.

Whilst interesting and worthy of publishing in Biogeosciences, I have some concerns with some of the quite sweeping conclusions that appear to be derived from comparison with a single dataset or single datapoint (referred to in more detail below). In addition, I have made other specific comments that I believe should be addressed before this manuscript is suitable for publication.

SPECIFIC COMMENTS

P 1, L27 – I am unclear whether the term '. . .real world Chl-a. . .' used here refers to actual in-situ chlorophyll-a concentration or satellite derived chlorophyll-a concentration. The term 'real' appears to be used interchangeably throughout the manuscript.

P5, L15 – Would it make more sense to swap Figures 1 and 2 so that the comparison of model and OC-CCI reflectance appears first as Fig 1 and is followed by the product comparisons as Fig 2. If this were the case, would there then be anything gained by comparing model actual and derived Chl-a to the OC-CCI Chl-a product? As I understand it, this paper is more about using the model output as a test ground to compare model 'actual' to 'derived' Chl-a rather than testing how well the model replicates the

'real world' values. Just a thought.

P6, L13 – The authors talk about comparing '...locations and dates similar to those in NOMAD.' What is their definition of similar?

P6, L15 – Again they use 'similar' to describe the resulting relationship between model 'actual' chlorophyll, model X, and real world in situ observations without really defining what similar means.

P6, L20 – The authors make no mention of the discrepancy between model reflectance wavebands (blue – 450 nm, 475 nm or 500 nm, green – 550 nm) and those used in the OC4 (blue – 443 nm, 490 nm or 510 nm, green – 555 nm) or OC3M-547 (blue – 443 nm or 488 nm, green – 547 nm) algorithms when comparing coefficients in Table 1. It might make it clearer to those not familiar with the derivation of these algorithms that they are not comparing like with like.

P7, L1-5 – The authors compare the OC-CCI Chl-a product to model derived Chl-a (although see my second comment). Where are the plots to support the statistics? What monthly climatologies are used to generate these statistics (is it a combination of Jan and Jul or all months?) Over what period are the January and July OC-CCI mean values determined? Are the OC-CCI output averaged to 1 degree by 1 degree similar to the model output? What version of the OC-CCI product is used? Are these OC-CCI products just OCx type output or do they include data from the Hu CI algorithm? The OC-CCI output is just one product. The statement at L5 seems to be quite a bold statement to make when only one product has been compared.

P7, L20 – Perhaps it's my eyesight but I'm not convinced that Figs (b) and (e) show '...much lower biases at high latitudes...'. (I assume you are comparing Fig 6 (b) and (e) to Fig. 6 (a) and (d))

P8, L3-5 – Are grid cells with depths less than 1000m also excluded?

P8, L22-24 – These statements appear to be derived from data taken from one point

in the North Atlantic. Is this a fair representation of the global pattern or is it just representative of this location?

P9, L4 – The authors could reference the Dutkiewicz et al. (2015) paper again here.

P9, L5 – The authors refer to 'studies' then reference a single instance.

P9, L17-21 How do the authors support this statement? If it is the timeseries data in Figure 8, then these are data from just one point in the North Atlantic. I don't think that data from one location and for one year is sufficient to warrant these conclusions.

Figure 3 – How do you differentiate between zero bias and lack of data? Could I suggest that lack of data is coloured differently to zero bias?

Figure 4 – Not sure whether the figure order works. The first mention of Fig. 4 that I can find occurs on P11 after reference to all the other figures. Again, if the authors are comparing the polynomials it might make it clearer to the reader if they acknowledge that different wavelengths have been used in the derivation in the legend.

Figure 8 – I don't think the x axis matches the label in Fig 8 (b). I assume the vertical dotted line marks the peak in 'actual' Chl-a?

TECHNICAL CORRECTIONS

P1, L16 – Should read '. . .Chl-a to the actual. . .'

P1, L25 – Should read 'This result indicates. . .'

P2, L15 – I think this is the first use of the acronym CDOM and so it should be defined here.

P2, L18 – Should read 'There have been. . .'

P2, L20 – Should read 'product' instead of 'products'

P2, L24 repeats L7

P3, L28 – In situ is italicised here but nowhere else.

P4, L14, 15, 17, 19 - Repeated uses of 'explicit'.

P5, L15 – 'Fig.2' is italicised

P7, L4 – Missing figure number

P7, L19 - Should read '. . .lead to a better. . .'

P9, L3 – Should read 'lead' rather than 'leads'

P9, L21 – Should read '. . .remains relatively high. . .'

P9, L29 – Don't think there should be a comma after 'pigments'.

P9, L31 – However, I think there should be one after 'reflectance'.

P12, L22 Should read '. . .by-products. . .'

---

## Author Comment (AC1) · 7 Nov 2017

We thank the reviewer for this constructive review, and respond to each point below in blue text, and proposed altered text in italics.

This manuscript provides an overview on the how a coupled biogeochemical-ecosystem-optical model can be used to explore ocean colour algorithms, with a focus on Chlorophyll-a. The authors effectively show the kind of interrogation studies that can be done with this type of "virtual laboratory". They clearly demonstrate how the ocean colour community can explore the bias and uncertainties of algorithms and their products, by investigating the effect of (1) other optically significant materials on derived Chlorophyll-a, and (2) different sized and regionally focused training datasets on robustness of an algorithm. I think this manuscript paves the ground for more detailed studies on the use of a radiative transfer component in a biogeochemical-ecosystem model to investigate ocean colour algorithms. The manuscript is well-written and logically presented, but there are a couple of points where I think a bit more clarity would improve the presentation of the methods & results (see comments).

We thank the reviewer for these positive comments, and especially the understanding of the premise of the paper as a "virtual laboratory". We hope that this study will allow for others that use such a laboratory for studying ocean colour algorithms and products.

Specific comments:
P2 L20-21: the band-ratio definitely used to be the most commonly used Chl-a algorithm for NASA, but they switched their "default" Chl-a to a merged approach of Hu et al. (2012) and the OCx type algorithms in Reprocessing 2014.0. I am not suggesting you redo your analysis using the band-difference algorithm (because as I understand it, the point in the paper is more to show the kind of analysis you can do with this type of "virtual laboratory", and dealing with multiple Chl-a algorithms might confuse matters - that being said, it would be an interesting task), but I think it might be worth acknowledging that the OCx algorithms are not the most common for NASA anymore.
Hu et al. (2012), J Geophys. Res., 117(C1). doi: 10.1029/2011jc007395

This is a good point. We had used OC4, because this was what was used on the OC-CCI Chl-a product that we show in Figure 1. We however note that the latest OC-CCI product has switched to a blend of OC3, OC4+CI, OC5. But the improvements in algorithms with new reprocessing of the NASA products is important to acknowledge. We plan to include statements to this effect in the introduction and conclusions of a revised paper. Including a statement that the model might be a good place to explore some of the newer algorithms.

In the introduction we now include the following (underlined are added and altered text):
*"There are significant work to improve algorithms. For instance, the newest National Aeronautics and Space Administration (NASA) reprocessing of Chl-a products has included a merged approach which uses different combination of reflectance bands at low Chl-a (Hu et al., 2012). There have also been many attempts to develop more mechanistically derived algorithms (e.g. using known relationships between absorption, scattering and reflectance). Here we focus on the Chl-a estimated from the blue/green reflectance as it is still the most commonly known product, and until very recently used in products downloaded from both NASA and the European Space Agency (ESA) data portals, as well as in merged products such as the Ocean Colour Climate Change Initiative (OC-CCI). However we note that similar techniques used in this paper could help inform on other algorithms. That the satellite-derived products have large errors and specific regional biases is relatively well understood in the ocean colour scientific*

*community (Hu et al., 2000, Moore et al., 2009; Blondeau-Parissier et al., 2014; Szeto et al., 2011). However, there remain many aspects of errors, biases and uncertainties that are poorly quantified, particularly…"*

In the conclusion, we finish with:

*"We also hope that the ocean colour community will see the potential of model approaches such as this for deriving sampling strategies, further studies on newer Chl-a algorithms (e.g. NASA Reprocessing 2014.0, and OC-CCI V3 release), other ocean colour products, and will help with algorithm developments for current and future ocean colour measurements."*

P4 L29-30: While this appears to be true for the January images, it seems to me that the July OC-CCI image (1d) has higher values in the northern high latitudes (around Greenland, Bering Sea, around Scandinavia) than actual July image (1e).

Yes, this is true. The larger values are most notable in the Southern Ocean and in January in the North Pacific. We will make this clearer in the revised version of the paper. We alter this sentence to:

*"As noted (and discussed more fully) in Dutkiewicz et al (2015) there are biases between the model and the observations, in particular larger values in the Southern Ocean and seasonally in the North Pacific than in the real-world satellite-derived Chl-a (Fig 1a,b,d,e)."*

We also are more precise when discussing the model Chl-a (both derived and actual) to OC-CCI at the end of section 3.1 (see below).

P5 L15-27: It is a bit unclear to me which results we are comparing at different points in this paragraph e.g. are the "observations" (L19) the OC-CCI observations? What is the "real world actual Chl-a" (L24)? L19-20: Are you saying the model blue Rrs is too high in the equatorial regions compared to the OC-CCI, coincident with where the model "actual" Chl-a is too low compared to the OC-CCI? Are you meaning OC-CCI is the "real ocean"? Maybe this sentence could be reworded to clarify this.

Indeed this is a difficult section to read (and write). We have tried to rewrite clearer, laying out the terminology first in the introduction. We now refer to the real world OC-CCI Chl-a product, rather suggesting OC-CCI is the "real ocean". We first have a statement in the introduction to emphasis why we use the term "real-world":

*""(In this article "real-world" will be used to refer to the real ocean and the real derived ocean colour products that are provide by space agencies. The "real world" is thus different to the numerical biogeochemical/ecosystem/optical model output and the products derived from it.)"*

And have rewritten this Rrs comparison section as (underlined is added or altered text):

*"We compare the model output to real world remotely sensed reflectance using the OC-CCI product (Fig 2). We note that the model does not have the exact same wavebands as any of the ocean colour satellites, and as such here we compare to the nearest bands: 450nm model to 443nm for the OC-CCI product, and 550nm model to the 555nm OC-CCI product. The model captures the reversed patterns between blue (443nm/450nm) and green (555nm/550nm) $R_{RS}$ between gyres and high productive regions. The model blue $R_{RS}$ (Fig 2a,b,c,d) captures the spatial and seasonal patterns in the real world satellite product. However, the model has lower blue $R_{RS}$ in the southern Pacific gyre in January. We note though that the model lowest Chl-a in this region is offset from the real-world OC-CCI product (Fig 1a,b). Similarly the model blue $R_{RS}$ is too high in the equatorial Atlantic and Pacific, but where the model Chl-a is likely too low relative to the real world Chl-a product (see Fig 1). The model has noticeably higher green (550nm) $R_{RS}$ in the equatorial Atlantic and Indian than the satellite measurements but note that these are regions of high cloud cover where the real world satellite product may be biased. We also find higher green $R_{RS}$ (Fig 2 e,f,g,h) in the North Pacific, but this might be due to model Chl-a being too high in this region (see Fig 1). In general the differences between model and the real world satellite $R_{RS}$ appear often to be linked to discrepancies between the model and real world satellite derived Chl-a product (and likely also in situ measurements). The model blue and green $R_{RS}$ appears to be consistent with the model actual Chl-a fields in a way that is similar to the real world and as such we believe appropriate and useful to use these model remotely sensed reflectance ("model ocean colour") to construct "satellite-like-derived" Chl-a using the blue to green reflectance ratio algorithm."*

P7 L4: I think this sentence could be more clearly explained. I think I understand the point you are making: that because the model derived Chl-a compares better with the OC-CCI Chl-a than the model actual Chl-a does, then some of the difference between OC-CCI and model actual Chl-a can be attributed to problems with the band-ratio algorithm (i.e. "product bias")? Is this what you mean by "product bias" - that there is an intrinsic problem with the band ratio formulation? I think the use of term "model" at the end of this sentence is particularly confusing: often in the ocean colour community, the term "model" is used in terms of a bio-optical proxy/relationship e.g. Chl-a is modelled using the band-ratio. Perhaps use "ecosystem model" (or something similar), to make this distinction clear.

Yes, you understand the point we are trying to make. We have rewritten this section to make this clearer, and in particular added "biogeochemical/ecosystem/optical" to clarify "model". This difference in use of "model" is important and we try to be careful not to be ambiguous in the paper. In particular we add a sentence to the last paragraph of the introduction to clarify this:

*"Additionally in this article, when we use the word "model", we refer to the numerical biogeochemical/ecosystem/optical model: In the ocean colour community "model" often refers to bio-optical relationships, we do not use "model" with this definition here."*

And this section (P7, L4, the last paragraph in Section 3.1) has been rewritten (taking Reviewer 2's comments into account as well) as (added or altered text underlined):

*"Finally in this section, we ask: Which model Chl-a (derived versus actual) best matches real-world OC-CCI product? We do not do this not for model validation purposes (see evaluation in Dutkiewicz et al., 2015), but rather to re-emphasis that the satellite derived Chl-a products are proxies for real world actual*

*Chl-a: the two are not the same thing. We compare model climatological monthly model derived Chl-a and model actual Chl-a to OC-CCI monthly climatology regridded to the model configuration (1 degree resolution). We find that the model derived Chl-a has global RMSE of 0.2867mg/m³, which is significantly lower than 0.6370mg/m³ found when comparing model actual Chl-a to OC-CCI. Comparisons are particularly better for the Southern Ocean and North Pacific (Fig 1). Consequently, some (though certainly not all) of the biases noted when comparing model actual Chl-a (Fig 1b,e) to real world satellite derived Chl-a products (Fig 1a,d, section 2 and in the model evaluation done in Dutkiewicz et al. 2015) are due to the real world Chl-a derived product bias and not a deficiency in the biogeochemical/ecosystem/optical model. It follows that a model satellite-like derived products (Fig 1 c,f) might be a better evaluation tools for comparing to ocean colour products derived with the same algorithm (Fig 1a,d) than the model actual Chl-a fields themselves."*

P10 L2-5 (& Appendix B): The exact method is a bit unclear to me here. Did you: take the results of the full run (i.e. those shown in Fig 5a), then take the monthly mean of Rrs output and Chl-a input and derive the 4th order polynomial coefficients on those monthly means? Then, for each of the 3 experiments, did you: do a full run with daily values, take the monthly mean of the model output Rrs and input Chl-a, and use the monthly band-ratio relationship with the monthly Rrs for input? Or did you set up the model with monthly means for the input? Perhaps this could be clarified in the text.

The problem with the sensitivity studies was saving the daily fields for 15 years. So instead we saved off the monthly fields instead. Thus model output for these sensitivity studies was monthly means of Rrs and monthly means of Chl-a. The "default" simulation was rerun saving off the monthly means (rather than the daily means used in Fig 5) so that it would be directly comparable to the sensitivity studies. The 4$^{th}$ order polynomials were calculated with these monthly means. We try to explain this clearer in the text now (underlined are altered text).

In main text we propose to have the following statement to make this clearer:

*"However given computational and storage constraints we used monthly averaged values of Chl-a and $R_{RS}$ to calculate the algorithm coefficients in these experiments rather than daily values (see Appendix B for discussion)."*

And for Appendix B we will suggest the following as a change (underlined are the altered text):

*"The daily values for 15 years, at each grid point creates a very large datafile. Diagnostics with, and storage of this large dataset becomes extremely computationally expensive. In order to conduct sensitivity studies we found that we needed to reduce this data set. Here we explore only outputing monthly means of model $R_{RS}$ and Chl-a and thus reducing the dataset by 1/30$^{th}$. We determined the algorithm coefficients ($a_0$ to $a_4$ in Eq 1) using monthly rather than daily means subsampling for the GS approach. The resulting function (Fig 4, solid black) is similar at low and intermediate of Chl-a, but does deviate at high Chl-a from the algorithm found using daily mean values (light blue line). The $r^2$ from this algorithm with coefficients defined with monthly means was also not quite as good as that found using daily means (see Table 2 and 3). However we found that the results were similar enough that we could obtain qualitative comparison between sensitivity experiments EXP-1, EXP-2, EXP-3 discussed in Section*

*5). We also note that the resulting two dimensional histogram (Fig 11) has far lower density when using 4 million relative to 140 million points. Though not perfect, using monthly output does allow us to perform EXP-1 through EXP-3 and still feel confident that the between experiment differences are robust."*

Fig 11: If I am understanding correctly, Fig 11a is the same as Fig 5a, but 11a uses the monthly coefficients i.e. the black line in Fig 4, whereas 5a uses the light blue line. I think it would be useful to point this out explicitly.

Yes, Fig 11a uses coefficients found using monthly mean Chl-a and R$_{RS}$. And yes, the different functions are given by light blue and black lines in Figure 4. We plan to add a further sentence to Fig 11 to make this point. Thus together with the sentences already in the caption we will have the following to clarify this different (underlined are altered text):

*"In these plots, monthly mean output of Chl-a and R$_{RS}$ were used to calculate the algorithm, and only monthly mean output is shown (4 million versus 140 million points), thus at a great computational savings. The difference in the algorithm is shown in Figure 4 (the light blue (coefficients using daily values) versus the solid black line (coefficients using monthly values). Differences between 11a and 5a are due to this difference in sampling (discussed in Appendix B). Also notice the difference in values on the colourbars between this figure and Figure 5."*

P12 L20-21 & L33-P13 L1: I'm not so sure it is quite as simple as this. I agree that the other optically significant materials are contributing to false Chl-a signals: there is a shoulder in all the derived Chl-a time series (Fig 8a), that aligns perfectly with the peak in CDOM and detritus (Fig 8b). But you can see the pattern of the actual Chl-a signal in the derived values, with peaks aligning on around days 60 and 75 - the magnitude of these derived values are just less than the actual values, but I'd say these are the "true peaks" of the spring bloom. After approximately day 75, there is the interference from CDOM and detritus, hence when calculating the initiation of the spring bloom (as described in the appendix), this large "false peak/shoulder" increases the median Chl-a and skews the determined initiation date. So I think what your data could be showing is (1) the Chl-a products do capture the peak of the spring bloom, but the magnitude of that peak is too small, and (2) the CDOM and detritus contribute to a false Chl-a signal, which makes it appear as if the bloom lasts longer and (depending on how you define bloom initiation) makes the initiation date appear to lag compared to the model actual.

We agree that that there is an alignment with the first peak (around day 60), but there is an offset for the maximum peak. Indeed, some of the points we make here are subject to what we define as the "initiation" and "peak", and we now make this clearer in the revised text: see below.
We use the definitions of Coles et al (2012) and to make our points clearer, we have added extra lines to Fig 8 (see below) to show the timing of the "initiation". Thus, for this definition of initiation there is indeed a lag between the derived products and the actual Chl-a. (Fig 9)
We add text to clarify this – i.e. acknowledging that the products are capturing the peaks at some times and not others. We also add a figure of the maximum peak offset as an additional panel for Figure 9 (b), as well as for the difference in timings of initiation for CDOM and detrital matter in response to a comment by reviewer 2.

Section 4 will now read (underlined indicates new or altered text):

*"We have noted that in all approaches, though even more obvious in RA, there is a seasonally altering pattern between the derived and actual model Chl-a (Fig 6). The amplitude of the peak of spring blooms is often underestimated in the products derived using global coefficients (GS and GA) in high latitude, especially in the subsampled algorithm (GS) (Figs 6). , Derived Chl-a values were also often higher than model actual Chl-a outside of bloom peaks. We consider the phenology using a single location (in the subpolar North Atlantic) for a single year as illustration (Fig 8a). Though the derived products show similar (though smaller) peaks to the actual Chl-a, and sometimes similar peak timing early in the season (see for instance the first distinct peak in this illustrative location), there are noticeable lags for the maximum peak (shown with a dotted line) and other mismatches later in the season. We also find that the bloom period lasts later into the year. The actual Chl-a also starts its sharp increase in spring (the initiation of the spring bloom, shown with dashed line) considerably before all three derived products (Fig 8a). We follow the approach of Cole et al (2012) for determining the "initiation of the spring bloom" as the time when the Chl-a first increases 5% above the annual median (horizontal dashed line, more description in Appendix A).*

*Figure 8 shows just one location for 1 year. To consider the large scale patterns, we determine the lag in the spring initiation (Fig 9a) and maximum bloom timing (Fig 9b) for each location averaged over all years. We find that in almost all locations the derived Chl-a shows the bloom starting later than the model actual Chl-a (Fig 9a). This offset is typically by about 5-10 days but can be as much as 30 days. The maximum Chl-a from the derived product also lags the actual Chl-a in most locations, though by only a few days (Fig 9b). These results indicate that temporal as well as spatial biases occur as a result of deriving Chl-a from X and suggests care should be taken when calculating phenology from satellite products or when evaluating phenology in models using satellite-derived Chl-a. We discuss the reason for the lags in the next section."*

And the following caveats in section 5:

*"We add the caveats that the exact definition of "initiation of bloom" does impact how much of a lag there is in the phenology. For instance, if the first peak in the model actual Chl-a in Figure 8a was defined as "the spring bloom" we would suggest the derived Chl-a does capture the timing better (though not the magnitude). We also note that the model parameterization of CDOM and detrital particle are not necessarily sufficiently well developed to make quantitative statements on the likely real-world lags. Thus though we do suggest there could be significant lags in phenology in the real world, we do not suggest that the values in Figure 9 are necessarily accurate for the real world. This analysis should instead be seen as a cautionary statement about using satellite-derived products for phenology of the quantities for which they are proxies."*

And more-over are careful in the rest of the text to discuss "phenology" rather than "spring bloom" such that the role of definitions of timing are not as relevant.

New versions of Figure 8 and 9 and their captions (altered of added text underline) are provide here. Extra panels in Figure 9 are added at Reviewer 2's suggestion.

[Figure]

*Figure 8: Illustrative timeseries for one year from a single location in the North Atlantic (shown as x on Fig 9). (a) "actual" Chl-a (black), derived Chl-a using subsampled output (GS, light blue), derived Chl-a using all output (GA, dark blue), and the Chl-a product derived using a regional specific algorithm (RA, purple). (b) actual Chl-a (black), CDOM (red) and detritus (green), all normalized to their peak value. Dashed vertical line indicates the "initiation of the bloom" which is taken to be when Chl-a reaches 5% above the annual median value (dotted horizontal line shows this value for the model actual Chl-a), following Cole et al (2012) and discussed further in Appendix A. The vertical dotted line indicates the peak of the bloom. Shown here is only a single year and location, however for larger scale perspective, the difference in initiation and peak timing between model actual and derived Chl-a averaged over all years are shown for the globe in Figure 9.*

[Figure]

*Figure 9: Lag in phenology. Number of days between a) the initiation of the spring bloom from model actual Chl-a and that for the model derived Chl-a (GS); b) yearly maximum of model actual Chl-a and that for the derived Chl-a (GS); c) initiation of the spring bloom from model actual Chl-a and the initiation of the CDOM increase; d) initiation of the spring bloom from model actual Chl-a and the initiation of detrital particle increase. Bloom initiation is defined as when Chl-a, CDOM or detrital particles reach 5% above their annual median value (see Appendix A). Lack of output indicate did regions with no significant seasonal cycle or are not resolved by the model (e.g. Arctic Ocean).*

It might be useful to have a table presenting the numerical results (e.g. log/linear RMSE, absolute % bias, etc.) for each approach in Section 3 and 5, to make it easier to compare the different results.

We now plan to add two tables (Table 2 and 3, shown below), one for each section. We split into two tables, to avoid comparison between the daily and the monthly determined algorithms. We also emphasis this difference in the table 3's caption

|  | Approach 1: GS | Approach 2: GA | Approach 3: RA |
|---|---|---|---|
| $r^2$ (log space) | 0.9088 | 0.9222 | 0.9466 |
| RMSE (log space) | 0.1599 | 0.1477 | 0.1215 |
| $r^2$ (linear space) | 0.6014 | 0.7670 | 0.8301 |
| RMSE (linear space) | 0.4816 | 0.3682 | 0.3083 |
| absolute % bias | 22% | 23% | 17% |

**Table 2: Results of comparison between model "actual" and model "satellite-like" derived Chl-a for the three algorithm approaches discussed in Section 3. Statistics are calculated for each grid, and each day**

*over 15 years, except for grid cells and times with low light, very low Chl-a and shallow regions (see text).*

| | Default | EXP-1: uniform $a_{CDOM}$ | EXP-2: uniform $a_{det}$ | EXP-3: phytoplankton optical same |
|---|---|---|---|---|
| $r^2$ (log space) | 0.8999 | 0.8742 | 0.8905 | 0.9493 |
| RMSE (log space) | 0.1678 | 0.1636 | 0.1663 | 0.1208 |
| $r^2$ (linear space) | 0.5373 | 0.6298 | 0.5991 | 0.7520 |
| RMSE (linear space) | 0.4420 | 0.3811 | 0.3962 | 0.2591 |
| absolute % bias | 21% | 20% | 23% | 18% |

*Table 3: Results of comparison between model "actual" and model "satellite-like" derived Chl-a for the sensitivity experiments discussed in Section 5. All "satellite-like" derived Chl-a was calculated using the GS approach. "Default" is the full experiment discussed in Section 3, but with monthly $R_{RS}$ used to calculate the algorithm coefficients. Statistics are calculated for each grid, and each month over 15 years, except for grid and times with low light, very low Chl-a and shallow regions (see text).*

Fig 4 and Fig 8: Legends would make these plots easier to read.

Agreed. We will add these in the revised paper (see version of Fig 8 above).

Fig 5 and Fig 11: it would be useful to have a title or text on each graph to show which subplot is which e.g. "(a) GS"

Agreed, we will add these in the revised paper.

Fig 8: The thick black line on top of the other time series signals masks some of the detail of the derived Chl-a products, particularly after the first 3 months - could this be represented in a different way? Also, check the axis labels, I think Fig 8b is showing days, not months.

We've redone the figure with a thinner black line, and place it under the other lines: the resulting figures is clearer (see above). And yes, Fig 8b x-axis was days, we now change so that Fig 8a and 8b have the same x-axis. Thank you for catching this.

Technical Corrections:
P1 L15-16: missing the word "to" i.e. sentence should read "...derived Chl-a to the actual..."

Thank you – added in the revised text.

P1 L25: should be either "These results indicate" or "This result indicates"

Thanks – will fix in the revised text as "These results indicate…."

P9 L29: Should this sentence not end with a question mark? i.e. ". . .community structure)?"

Yes, will fix in revised text.

P11 L23: remove the second "like"

Thanks, will do

P12 L24: build should be built

Yes, thank you. Will fix in revised text.

---

## Author Comment (AC2) · 7 Nov 2017

We thank the reviewer for their comments, and respond to each point below in blue text, and proposed altered text in italics.

The paper 'Modelling Ocean Colour Derived Chlorophyll-a' by Dutkiewicz et al. uses a modified version of the MIT general circulation model that incorporates scattering and absorption properties of water, detritus, coloured dissolved organic matter (CDOM) and 9 phytoplankton types. 'Actual' chlorophyll-a concentration (Chl-a) is determined by summing the variable chlorophyll-a concentration of the 9 phytoplankton types. Remote sensing spectral reflectance, determined from the resulting modelled upwelling and downwelling irradiances are used in a NASA OCx type algorithm to compute satellite-like 'derived' chlorophyll-a concentration which is compared to 'actual' chlorophyll-a concentration. Firstly, the model output is used to test assumptions used in the derivation of chlorophyll-a concentration using the standard OCx ratio algorithm. Secondly, bloom initiation timings, determined from the 'derived' and 'actual' chlorophyll concenrations, are compared. Lastly, the impact that other optically important parameters may have on 'derived' chlorophyll concentration is explored. The authors conclude that (a) applying a single set of coefficients in the OCx algorithm globally is not as accurate as applying regionally variable coefficients, (b) there is a temporal mismatch between the initiation of the spring bloom defined using 'actual' Chl-a compared with that de-fined using 'derived' Chl-a and (c) that this mismatch may be caused by the optical influence of other substances such as CDOM and/or detritus.

I found this paper to be generally well-thought out and well-written. For colleagues who are not regular users of these products, this paper provides important caveats for the use of satellite derived data. For those of us who work with ocean colour derived products more often, the results may not be unsurprising, but it serves as a timely reminder of the limitations associated with satellite derived data.

We thank the reviewer for these positive comments and for recognizing that though some of the results are not surprising to those who are experts in ocean colour products, these results offer a reminder of the limitations. We do believe that for non-experts of ocean colour products, these results will be very informative.

Whilst interesting and worthy of publishing in Biogeosciences, I have some concerns with some of the quite sweeping conclusions that appear to be derived from comparison with a single dataset or single datapoint (referred to in more detail below). In addition, I have made other specific comments that I believe should be addressed be-fore this manuscript is suitable for publication.

The reviewer's comment relates to Fig 8, and the discussion of the results shown in it for one location and one year. This figure was only supposed to be illustrative, not the main basis for our conclusions. We have significantly rewritten this section to make this obvious, and include several more panels in Fig 9 that show that the inferences we gain from that location are relevant for much of the globe. See below.

SPECIFIC COMMENTS

P 1, L27 – I am unclear whether the term '. . .real world Chl-a. . .' used here refers to actual in-situ chlorophyll-a concentration or satellite derived chlorophyll-a concentration. The term 'real' appears to be used interchangeably throughout the manuscript.

We agree that this incident of the use of "real" was confusing and will change this to be "to real world satellite-derived Chl". We have also gone through the manuscript being careful how we use "real world" in the text. We also add a sentence in the introduction to define "real world" for the rest of the paper:

"*"(In this article "real-world" will be used to refer to the real ocean and the real derived ocean colour products that are provide by space agencies. The "real world" is thus different to the numerical biogeochemical/ecosystem/optical model output and the products derived from it.)"*

P5, L15 – Would it make more sense to swap Figures 1 and 2 so that the comparison of model and OC-CCI reflectance appears first as Fig 1 and is followed by the product comparisons as Fig 2. If this were the case, would there then be anything gained by comparing model actual and derived Chl-a to the OC-CCI Chl-a product? As I under-stand it, this paper is more about using the model output as a test ground to compare model 'actual' to 'derived' Chl-a rather than testing how well the model replicates the real world' values. Just a thought.

We like the "thought" – it would make a cleaner paper not getting into the aspect of evaluating the model. However, one of the points we would like to make is that comparing model "actual" or model "derived" Chl-a to "real-world" satellite derived Chl-a leads to different results. There are therefore important implications of our analysis for model validation (i.e. that comparing model chl-a to real-world satellite-derived Chla is not comparing like-for-like), so we feel this comparison is an important part of the paper. We have considerably rewritten the relevant paragraph (last one of section 3.1) to make this point clearer and to emphasis that this is not a "evaluation", but a demonstration (underlined is added or altered text).

*"Finally in this section, we ask: Which model Chl-a (derived versus actual) best matches real-world satellite-like derived Chl-a (here the OC-CCI product)? We do not do this not for model validation purposes (see evaluation in Dutkiewicz et al., 2015), but rather to re-emphasis that the derived Chl-a product is only a proxy for actual Chl-a: the two are not the same thing. We compare model climatological monthly model derived Chl-a and model actual Chl-a to OC-CCI monthly climatology regridded to the model configuration (1 degree resolution). We find that the model derived Chl-a has global RMSE of 0.2867mg/m³, which is significantly lower than 0.6370mg/m³ found when comparing model actual Chl-a to OC-CCI. Comparisons are particularly better for the Southern Ocean and North Pacific (Fig 1). Consequently, some (though certainly not all) of the biases noted when comparing model actual Chl-a (Fig 1b,e) to real world satellite derived Chl-a products (Fig 1a,d, section 2 and in the model evaluation done in Dutkiewicz et al. 2015) are due to the real world Chl-a derived product bias and not a deficiency in the biogeochemical/ecosystem/optical model. It follows that a model satellite-like derived products (Fig 1 c,f) might be a better evaluation tools for comparing to ocean colour products derived with the same algorithm (Fig 1a,d) than the model actual Chl-a fields themselves."*

Though we agree that it would be nice to have figure 2 (reflectance) before figure 1 (product: Chl-a) we could not find a way to do this without complicating the paper. So we have left as is.

P6, L13 – The authors talk about comparing '. . .locations and dates similar to those in NOMAD.' What is their definition of similar?

The model has 1 degree resolution – so we are using the degree box within which the actual is situ measurements are taken. This is unnecessarily confusing though and we now state differently:

*".. at locations and dates nearest those in NOMAD"*

P6, L15 – Again they use 'similar' to describe the resulting relationship between model 'actual' chlorophyll, model X, and real world in situ observations without really defining what similar means.

Here we had meant to reference Figure 4, where we show the OC4 and OC3M-547 functions. We now change this sentence to reflect this:

*"The resulting relationship between model blue/green reflectance ratio (X) and Chl-a from subsampling the model (Fig 3a) is similar to that found for real-world algorithms (Fig 4)."*

P6, L20 – The authors make no mention of the discrepancy between model reflectance wavebands (blue – 450 nm, 475 nm or 500 nm, green – 550 nm) and those used in the OC4 (blue – 443 nm, 490 nm or 510 nm, green – 555 nm) or OC3M-547 (blue – 443 nm or 488 nm, green – 547 nm) algorithms when comparing coefficients in Table 1. It might make it clearer to those not familiar with the derivation of these algorithms that they are not comparing like with like.

Yes, thank you. This is an important point and in the revised version we will add the following to the text and to the figure caption.

We will add where we compare the model RRS to OC-CCI Rrs:

*"We note that the model does not have the exact same wavebands as any of the ocean colour satellites, and as such here we compare to the nearest bands: 450nm model to 443nm for the OC-CCI product, and 550nm model to the 555nm OC-CCI product."*

Also when we compare the model GS coefficients to OC4 and OC3-457 (Fig 4 and Table 1) we now add the following sentence:

*"Some of the differences between real-world and model coefficients is likely to come from the use of different exact bands in the blue and green (e.g. 550nm for model green versus 555nm for OC-CCI)."*

We also add the following to the figure caption in Figure 2:

*"We compare the model wavebands against the nearest OC-CCI wavebands, but note that they are not identical."*

P7, L1-5 – The authors compare the OC-CCI Chl-a product to model derived Chl-a (although see my second comment). Where are the plots to support the statistics? What monthly climatologies are used to generate these statistics (is it a combination of Jan and Jul or all months?) Over what period are the January and July OC-CCI mean values determined? Are the OC-CCI output averaged to 1 degree by 1 degree similar to the model output? What version of the OC-CCI product is used? Are these OC-CCI products just OCx type output or do they include data from the Hu CI algorithm? The OC-CCI output is just one product. The statement at L5 seems to be quite a bold statement to make when only one product has been compared.

The product from OC-CCI was OC4 without the Hu CI component. We discuss the Hu et al adjustments now in the introduction and text (see comments by Reviewer 1).

The RMSE is determined from daily values over the full time period (not just Jan and Jul) averaged globally. We have regridded OC-CCI onto the same grid as the model for these comparisons.

We would prefer not to add any additional figures, especially for this point. However, we do agree with the reviewers' concern on this section and have considerably rewritten it. We add some additional text (also in the figure caption) to explain the statistics better and state what OC-CCI product we are using. Additionally we add text in both introduction and summary to highlight that OC4 is just one product, and that there are newer (and better) products also available. We have stuck here to the OC4 as it was the one that compared to the version of OC-CCI that we downloaded (a new version (V3) has just been released). We do specifically mention the newer OCx+CI versions. We are not convinced that the statement on L5 is too bold. However we do add a caveat in it.

The paragraph questioned here is provided above for the discussion on Fig 1 (P5, L15) above. And we add the following sentence to caption of Figure 1:

*"We use version 2 of the OC-CCI, which uses an OC4 algorithm for determining the Chl-a product, and thus comparable algorithm as used in our model derived Chl-a shown in e,f."*

In the introduction we have the following (underlined is added or altered text):

*"There is significant ongoing work to improve algorithms. For instance, the newest National Aeronautics and Space Administration (NASA) reprocessing of Chl-a products has included a merged approach which uses different combination of reflectance bands at low Chl-a (Hu et al., 2012). There have also been many attempts to develop more mechanistically derived algorithms (e.g. using known relationships between absorption, scattering and reflectance). Here we focus on the Chl-a estimated from the blue/green reflectance as it is still the most commonly known product, and until very recently used in products downloaded from both NASA and the European Space Agency (ESA) data portals, as well as merged products such as the Ocean Colour Climate Change Initiative (OC-CCI). However we note that similar techniques used in this paper could help inform on other algorithms."*

The final sentence of the summary now reads (underlined is new text):

*"We also hope that the ocean colour community will see the potential of model approaches such as this for deriving sampling strategies, further studies on newer Chl-a algorithms (e.g. NASA Reprocessing 2014.0, and OC-CCI V3 release), other ocean colour products, and will help with algorithm developments for current and future ocean colour measurements."*

P7, L20 – Perhaps it's my eyesight but I'm not convinced that Figs (b) and (e) show '. . .much lower biases at high latitudes. . .'. (I assume you are comparing Fig 6 (b) and (e) to Fig. 6 (a) and (d))

Yes we were comparing b,e to a,d – will make clearer in the revised version. And we agree that this is not obvious from the figure – the improvement is mostly just at the edge of the data before moving into the white areas. We now restate this sentence (underlined is new/altered text):

*"Though there is some improvements in some regions in the higher latitudes, there is actually decrease in skill at lower latitudes (Fig 6b,e compared to a,d). There is in fact a slight increase in the mean % absolute bias (23%) between this and the GS estimates: When transformed into percent errors the increased biases at low Chl-a, low latitude regions become more prominent."*

P8, L3-5 – Are grid cells with depths less than 1000m also excluded?

Yes, will be stated in the revised version.

P8, L22-24 – These statements appear to be derived from data taken from one point in the North Atlantic. Is this a fair representation of the global pattern or is it just representative of this location?

We use the one location as an illustration. The results shown in figure 9 show that same lag in initiation of the bloom occur across much of the high latitudes. We make clear in the text and Figure 8 caption that this is just an illustration. To further bolster the relevance of this discussion in space and time, we now include other panels in figure 9 to show the mismatch in the peak timing, as well as in the initiation of increase in CDOM and detrital matter relative to actual Chl-a.

Section 4 now reads (underlined is altered or added text):

*"We have noted that in all approaches, though even more obvious in RA, there is a seasonally altering pattern between the derived and actual model Chl-a (Fig 6). The amplitude of the peak of spring blooms is often underestimated in the products derived using global coefficients (GS and GA) in high latitude, especially in the subsampled algorithm (GS) (Figs 6). , Derived Chl-a values were also often higher than model actual Chl-a outside of bloom peaks. We consider the phenology here, using a single location (in the subpolar North Atlantic) for a single year as illustration (Fig 8a). Though the derived products show similar (though smaller) peaks to the actual Chl-a, and sometimes similar peak timing early in the season (see for instance the first distinct peak in this illustrative location), there are noticeable lags for the maximum peak (shown with a dotted line) and other mismatches later in the season. We also find that*

*the bloom period last later into the year. The actual Chl-a also starts its sharp increase in spring (the initiation of the spring bloom, shown with dashed line) considerable before all three derived products (Fig 8a).  We follow the approach of Cole et al (2012) for determining the "initiation of the spring bloom" as the time when the Chl-a first increases 5% above the annual median (horizontal dashed line, more description in Appendix A).*

*Figure 8 shows just one location for 1 year. To consider the large scale patterns, we determine the lag in the spring initiation (Fig 9a) and maximum bloom timing (Fig 9b) for each location averaged over all years. We find that in almost all locations the derived Chl-a shows the bloom starting later than the model actual Chl-a (Fig 9a). This offset is typically by about 5-10 days but can be as much as 30 days. The maximum Chl-a from the derived product also lags the actual Chl-a in most locations, though by only a few days (Fig 9b). These result indicates that temporal as well as spatial biases occur as a result of deriving Chl-a from X and suggests care should be taken when calculating phenology from satellite products or when evaluating phenology in models using satellite-derived Chl-a. We discuss the reason for the lags in the next section."*

Figure 8 is now more clearly described as just an illustration with the connection to Figure 9 made to make obvious that Fig 9 has the global results. We also add additional lines to Fig 8 to show the initiation of the bloom. We attach the new Fig 8 and 9, and captions below.

[Figure]

*Figure 8: Illustrative timeseries for one year from a single location in the North Atlantic (shown as x on Fig 9). (a) "actual" Chl-a (black), derived Chl-a using subsampled output (GS, light blue), derived Chl-a using all output (GA, dark blue), and the Chl-a product derived using a regional specific algorithm (RA, purple). (b) actual Chl-a (black), CDOM (red) and detritus (green), all normalized to their peak value. Dashed vertical line indicates the "initiation of the bloom" which is taken to be when Chl-a reaches 5% above the annual median value (dotted horizontal line shows this value for the model actual Chl-a), following Cole et al (2012) and discussed further in Appendix A. The vertical dotted line indicates the peak of the bloom. Shown here is only a single year and location, however for larger scale perspective, the difference in initiation and peak timing between model actual and derived Chl-a averaged over all years are shown for the globe in Figure 9.*

[Figure]

*Figure 9: Lag in phenology. Number of days between a) the initiation of the spring bloom from model actual Chl-a and that for the model derived Chl-a (GS) ;b) yearly maximum of model actual Chl-a and that for the derived Chl-a (GS); c) initiation of the spring bloom from model actual Chl-a and the initiation of the CDOM increase; d) initiation of the spring bloom from model actual Chl-a and the initiation of detrital particle increase. Bloom initiation is defined as when Chl-a, CDOM or detrital particles reach 5% above their annual median value (see Appendix A). Lack of output indicate did regions with no significant seasonal cycle or are not resolved by the model (e.g. Arctic Ocean).*

P9, L4 – The authors could reference the Dutkiewicz et al. (2015) paper again here.

Good idea, we will do so in the revised version.

P9, L5 – The authors refer to 'studies' then reference a single instance.

We add "e.g" to the text and additional references (e.g. Loisel et al., 2010; Brown et al., 2008, Siegel et al., 2005a, 2005b)

Brown, C.A., Huot, Y., Werdell, P.J., Gentili, B., and Claustre, H.: The origin and global distribution of second order variability in satellite ocean color and its potential applications to algorithm development, Remote Sensing of Environment, 112, 4186-4203.

Loisel, H., Lebac, B., Dessailly, D., Duforet-Gaurier, L., and Vantrpotte, V.: Effect of inherent optical properties variability on chlorophyll retrieval from ocean colr remote sensing: an in situ approach. Optics Express, 18,

Siegel, D.A., Maritorena, S. Nelson, N.B., and Behrenfeld, M.J.: Independence and interdependencies of global ocean color properties; Reassessing the bio-optical assumption. J. Geophys. Res., 110, C07011, doi:10.1029/2004JC002527, 2005a

Siegel, D.A., Maritorena, S., Nelson, N.B., Behrenfeld, M.J. and McClain, C.R.: Colored dissolved organic matter and its influence on the satellite-based characterization of the ocean biosphere, Geophys. Res. Letters, 32, L20605, doi:10.1029/2005GL024310, 2005b.

P9, L17-21 How do the authors support this statement? If it is the timeseries data in Figure 8, then these are data from just one point in the North Atlantic. I don't think that data from one location and for one year is sufficient to warrant these conclusions.

See our comments above for P8, L22-24 above. The results in Figure 9 show the global results averaged over 13 years. However we do see the reviewers point, and now add the extra panels (c,d) to Figure 9. We modify the text in Section 5 (original P9, L17-19) as such (underlined is added or altered text):

*"However, we find that though linked, there are noticeable lags in the sharp increase in accumulation (Fig 8b, Fig 9 c,d) and peak timing and decline(Fig 8b) between CDOM and detrital matter and the model actual Chl-a."*

Figure 3 – How do you differentiate between zero bias and lack of data? Could I suggest that lack of data is coloured differently to zero bias?

I assume you mean Figure 6? Good idea. We will do so in the revised paper.

Figure 4 – Not sure whether the figure order works. The first mention of Fig. 4 that I can find occurs on P11 after reference to all the other figures. Again, if the authors are comparing the polynomials it might make it clearer to the reader if they acknowledge that different wavelengths have been used in the derivation in the legend.

Figure 4 is mentioned first on pg 6 (line 22), but we now also reference it earlier in the newer version of the paper. Since two of lines in Figure 4 are those in Figure 3, it does quite naturally belong here.

We add the acknowledgment of different wavelengths in the caption of Figure 4:
*"Note that the algorithms for the model come from band ratio of 425nm/450nm/475nm and 550nm. For the real world algorithms the band ratios are different and specific for the satellite sensor (SeaWifs or MODIS)."*

Figure 8 – I don't think the x axis matches the label in Fig 8 (b). I assume the vertical dotted line marks the peak in 'actual' Chl-a?

Thank you for catching this – we have altered the axis to match Fig 8a. Yes, the vertical dotted line is the peak – we now mention this in the figure caption. We have also added a line to show the initiation of the spring bloom for clarity. See new figure 8 and caption above.

TECHNICAL CORRECTIONS

P1, L16 – Should read '. . .Chl-a to the actual. . .'

Thank you – we will fix this.

P1, L25 – Should read 'This result indicates. . .'

Actually, I think should read "These results indicate…", so will change to this instead. Thank you for catching this inconsistency.

P2, L15 – I think this is the first use of the acronym CDOM and so it should be defined here.

Yes, will add.

P2, L18 – Should read 'There have been. . .'

Yes, thank you.

P2, L20 – Should read 'product' instead of 'products'

Yes, will fix

P2, L24 repeats L7

We will remove the text at L24 from the revised text.

P3, L28 – In situ is italicised here but nowhere else.

Thanks – we will remove the italics to be consistent.

P4, L14, 15, 17, 19 - Repeated uses of 'explicit'.

Yes, will redo to remove excessive use of "explicit". Thank you.

P5, L15 – 'Fig.2' is italicized

We will remove the italics. Thank you.

P7, L4 – Missing figure number

Will correct.

P7, L19 - Should read '. . .lead to a better. . .'

Yes, thank you.

P9, L3 – Should read 'lead' rather than 'leads'

Yes, thank you

P9, L21 – Should read '. . .remains relatively high. . .'

Yes, will fix

P9, L29 – Don't think there should be a comma after 'pigments'.

Indeed not, we have removed.

P9, L31 – However, I think there should be one after 'reflectance'.

Yes, have fixed this in revised version.

P12, L22 Should read '. . .by-products. . .'

Yes, thank you.

---

## Author Response (ED1)

Dear Emmanuel –

We have revised the manuscript following the reviewers' suggestions. We agree that the clarification based on the reviewer's comments leads to a much improved paper. We attach a version with all changes shown (using track changes in Word) along with the response to reviewers, as well as a version without the track changes.

The responses to the reviewers is almost identical to that already online in the discussion, except that we have now added line number for revisions specific for the track-changes version of the revised manuscript, and corrected a few minor typos and grammatical issues.

In response to the reviews we have added two tables (Table 2 and 3), and added extra panels to Fig 9. We have also redone Fig 4,5,6, 8 and 11 for clarity.

Thank you very much for handling this paper and for earlier comments which also helped in the clarification and suitability to the community.

Yours,

Stephanie Dutkiewicz

We thank the reviewer for this constructive review, and respond to each point below in blue text, and altered text in italics.

This manuscript provides an overview on the how a coupled biogeochemical-ecosystem-optical model can be used to explore ocean colour algorithms, with a focus on Chlorophyll-a. The authors effectively show the kind of interrogation studies that can be done with this type of "virtual laboratory". They clearly demonstrate how the ocean colour community can explore the bias and uncertainties of algorithms and their products, by investigating the effect of (1) other optically significant materials on derived Chlorophyll-a, and (2) different sized and regionally focused training datasets on robustness of an algorithm. I think this manuscript paves the ground for more detailed studies on the use of a radiative transfer component in a biogeochemical-ecosystem model to investigate ocean colour algorithms. The manuscript is well-written and logically presented, but there are a couple of points where I think a bit more clarity would improve the presentation of the methods & results (see comments).

We thank the reviewer for these positive comments, and especially the understanding of the premise of the paper as a "virtual laboratory". We hope that this study will allow for others that use such a laboratory for studying ocean colour algorithms and products.

Specific comments:
P2 L20-21: the band-ratio definitely used to be the most commonly used Chl-a algorithm for NASA, but they switched their "default" Chl-a to a merged approach of Hu et al. (2012) and the OCx type algorithms in Reprocessing 2014.0. I am not suggesting you redo your analysis using the band-difference algorithm (because as I understand it, the point in the paper is more to show the kind of analysis you can do with this type of "virtual laboratory", and dealing with multiple Chl-a algorithms might confuse matters - that being said, it would be an interesting task), but I think it might be worth acknowledging that the OCx algorithms are not the most common for NASA anymore.
Hu et al. (2012), J Geophys. Res., 117(C1). doi: 10.1029/2011jc007395

This is a good point. We had used OC4, because this was what was used on the OC-CCI Chl-a product that we show in Figure 1. We however note that the latest OC-CCI product has switched to a blend of OC3, OC4+CI, OC5. But the improvements in algorithms with new reprocessing of the NASA products is important to acknowledge. We plan to include statements to this effect in the introduction and conclusions of a revised paper. Including a statement that the model might be a good place to explore some of the newer algorithms.

In the introduction (pg 2, line19-30) we now include the following (underlined are added and altered text):
""*There is significant ongoing work to improve algorithms. For instance, the newest National Aeronautics and Space Administration (NASA) reprocessing of Chl-a products has included a merged approach that uses different combination of reflectance bands at low Chl-a (Hu et al., 2012). There have also been many attempts to develop more mechanistically derived algorithms (e.g. using known relationships between absorption, scattering and reflectance). Here we focus on the Chl-a estimated from the blue/green reflectance as it is still the most commonly known product, and until very recently used in products downloaded from both NASA and the European Space Agency (ESA) data portals, as well as merged products such as the Ocean Colour Climate Change Initiative (OC-CCI). However we note that similar techniques used in this paper could help inform on other algorithms. That the satellite-derived*

*products have large errors and specific regional biases is relatively well understood in the ocean colour scientific community (Hu et al., 2000, Moore et al., 2009; Blondeau-Parissier et al., 2014; Szeto et al., 2011). However, there remain many aspects of errors, biases and uncertainties that are poorly quantified, particularly…"*

In the conclusion (pg 15, lines 26-29), we finish with:

*"We also hope that the ocean colour community will see the potential of model approaches such as this for deriving sampling strategies, further studies on newer Chl-a algorithms (e.g. NASA Reprocessing 2014.0, and OC-CCI V3 release), other ocean colour products, and will help with algorithm developments for current and future ocean colour measurements."*

P4 L29-30: While this appears to be true for the January images, it seems to me that the July OC-CCI image (1d) has higher values in the northern high latitudes (around Greenland, Bering Sea, around Scandinavia) than actual July image (1e).

Yes, this is true. The larger values are most notable in the Southern Ocean and in January in the North Pacific. We will make this clearer in the revised version of the paper. We alter this sentence (pg 5, line 7-9) to:

*"As noted (and discussed more fully) in Dutkiewicz et al (2015) there are biases between the model and the observations, in particular larger values in the Southern Ocean and seasonally in the North Pacific than in the real-world satellite-derived Chl-a (Fig 1a,b,d,e)."*

We also are more precise when discussing the model Chl-a (both derived and actual) to OC-CCI at the end of section 3.1 (see below).

P5 L15-27: It is a bit unclear to me which results we are comparing at different points in this paragraph e.g. are the "observations" (L19) the OC-CCI observations? What is the "real world actual Chl-a" (L24)? L19-20: Are you saying the model blue Rrs is too high in the equatorial regions compared to the OC-CCI, coincident with where the model "actual" Chl-a is too low compared to the OC-CCI? Are you meaning OC-CCI is the "real ocean"? Maybe this sentence could be reworded to clarify this.

Indeed this is a difficult section to read (and write). We have tried to rewrite clearer, laying out the terminology first in the introduction. We now refer to the real world OC-CCI Chl-a product, rather suggesting OC-CCI is the "real ocean". We first have a statement in the introduction (pg 3, line 26-28) to emphasis why we use the term "real-world":

*""(In this article "real-world" will be used to refer to the real ocean and the real derived ocean colour products that are provide by space agencies. The "real world" is thus different to the numerical biogeochemical/ecosystem/optical model output and the products derived from it.)"*

And have rewritten this Rrs comparison section (pg 5 line 27 to pg 6 line 9) as (underlined is added or altered text):

*"We compare the model output to real world remotely sensed reflectance using the OC-CCI product (Fig 2). We note that the model does not have the exact same wavebands as any of the ocean colour satellites, and as such here we compare to the nearest bands: 450nm model to 443nm for the OC-CCI product, and 550nm model to the 555nm OC-CCI product. The model captures the reversed patterns between blue (443nm/450nm) and green (555nm/550nm) $R_{RS}$ between gyres and high productive regions. The model blue $R_{RS}$ (Fig 2a,b,c,d) captures the spatial and seasonal patterns in the real world satellite product. However, the model has lower blue $R_{RS}$ in the southern Pacific gyre in January. We note though that the model lowest Chl-a in this region is offset from the real-world OC-CCI product (Fig 1a,b). Similarly the model blue $R_{RS}$ is too high in the equatorial Atlantic and Pacific, but where the model Chl-a is likely too low relative to the real world Chl-a product (see Fig 1). The model has noticeably higher green (550nm) $R_{RS}$ in the equatorial Atlantic and Indian than the satellite measurements but note that these are regions of high cloud cover where the real world satellite product may be biased. We also find higher green $R_{RS}$ (Fig 2 e,f,g,h) in the North Pacific, but this might be due to model Chl-a being too high in this region (see Fig 1). In general the differences between model and the real world satellite $R_{RS}$ appear often to be linked to discrepancies between the model and real world satellite derived Chl-a product (and likely also in situ measurements). The model blue and green $R_{RS}$ appears to be consistent with the model actual Chl-a fields in a way that is similar to the real world and as such we believe appropriate and useful to use these model remotely sensed reflectance ("model ocean colour") to construct "satellite-like-derived" Chl-a using the blue to green reflectance ratio algorithm."*

P7 L4: I think this sentence could be more clearly explained. I think I understand the point you are making: that because the model derived Chl-a compares better with the OC-CCI Chl-a than the model actual Chl-a does, then some of the difference between OC-CCI and model actual Chl-a can be attributed to problems with the band-ratio algorithm (i.e. "product bias")? Is this what you mean by "product bias" - that there is an intrinsic problem with the band ratio formulation? I think the use of term "model" at the end of this sentence is particularly confusing: often in the ocean colour community, the term "model" is used in terms of a bio-optical proxy/relationship e.g. Chl-a is modelled using the band-ratio. Perhaps use "ecosystem model" (or something similar), to make this distinction clear.

Yes, you understand the point we are trying to make. We have rewritten this section to make this clearer, and in particular added "biogeochemical/ecosystem/optical" to clarify "model". This difference in use of "model" is important and we try to be careful not to be ambiguous in the paper. In particular we add a sentence to the last paragraph of the introduction (pg 3, lines 29-31) to clarify this:

*"Additionally when we use the word "model" in this article, we refer to the numerical biogeochemical/ecosystem/optical model: In the ocean colour community "model" often refers to bio-optical relationships, we do not use "model" with this definition here."*

And this section (pg 7, lines 13-31, the last paragraph in Section 3.1) has been rewritten (taking Reviewer 2's comments into account as well) as (added or altered text underlined):

*"Finally in this section, we ask: Which model Chl-a (derived versus actual) best matches real-world OC-CCI product? We do not do this not for model validation purposes (see evaluation in Dutkiewicz et al., 2015), but rather to re-emphasis that the satellite derived Chl-a products are proxies for real world actual Chl-a: The two are not the same thing. We compare climatological monthly model derived Chl-a and model actual Chl-a to OC-CCI monthly climatology regridded to the model configuration (1 degree resolution). We find that the model derived Chl-a has global RMSE of 0.2867mg/m³, which is significantly lower than 0.6370mg/m³ found when comparing model actual Chl-a to OC-CCI. Comparisons are particularly better for the Southern Ocean and North Pacific (Fig 1). Consequently, some (though certainly not all) of the biases noted when comparing model actual Chl-a (Fig 1b,e) to real world satellite derived Chl-a products (Fig 1a,d, section 2 and in the model evaluation done in Dutkiewicz et al. 2015) are due to the real world Chl-a derived product bias and not a deficiency in the biogeochemical/ecosystem/optical model. It follows that a model satellite-like derived products (Fig 1 c,f) might be a better evaluation tool for comparing to ocean colour products derived with the same algorithm (Fig 1a,d) than the model actual Chl-a fields themselves."*

P10 L2-5 (& Appendix B): The exact method is a bit unclear to me here. Did you: take the results of the full run (i.e. those shown in Fig 5a), then take the monthly mean of Rrs output and Chl-a input and derive the 4th order polynomial coefficients on those monthly means? Then, for each of the 3 experiments, did you: do a full run with daily values, take the monthly mean of the model output Rrs and input Chl-a, and use the monthly band-ratio relationship with the monthly Rrs for input? Or did you set up the model with monthly means for the input? Perhaps this could be clarified in the text.

The problem with the sensitivity studies was saving the daily fields for 15 years. So instead we saved off the monthly fields instead. Thus model output for these sensitivity studies was monthly means of Rrs and monthly means of Chl-a. The "default" simulation was rerun saving off the monthly means (rather than the daily means used in Fig 5) so that it would be directly comparable to the sensitivity studies. The 4th order polynomials were calculated with these monthly means. We try to explain this clearer in the text now (underlined are altered text).

In main text we propose to have the following statement (pg 11, lines 20-24) to make this clearer:

*"However, given computational and storage constraints we used monthly averaged values of Chl-a and $R_{RS}$ to calculate the algorithm coefficients in these experiments rather than daily values (see Appendix B for discussion)."*

And for Appendix B (pg 16) we will suggest the following as a change (underlined are the altered text):

*"The daily values for 15 years at each grid point creates a very large datafile. Diagnostics with, and storage of, this large dataset becomes extremely computationally expensive. In order to conduct sensitivity studies we found that we needed to reduce this data set. Here we explore only outputting*

*monthly means of model $R_{RS}$ and Chl-a and thus reducing the dataset by $1/30^{th}$. We determined the algorithm coefficients ($a_0$ to $a_4$ in Eq 1) using monthly rather than daily means and subsampling for the GS approach. The resulting function (Fig 4, solid black) is similar at low and intermediate Chl-a, but does deviate at high Chl-a from the algorithm found using daily mean values (light blue line). The $r^2$ from this algorithm with coefficients defined with monthly means was also not quite as good as that found using daily means (see Table 2 and 3). However we found that the results were similar enough that we could obtain qualitative comparison between sensitivity experiments EXP-1, EXP-2, EXP-3 discussed in Section 5). We also note that the resulting two dimensional histogram (Fig 11) has far lower density when using 4 million relative to 140 million points. Though not perfect, using monthly output does allow us to perform EXP-1 through EXP-3 and still feel confident that the between experiment differences are robust."*

Fig 11: If I am understanding correctly, Fig 11a is the same as Fig 5a, but 11a uses the monthly coefficients i.e. the black line in Fig 4, whereas 5a uses the light blue line. I think it would be useful to point this out explicitly.

Yes, Fig 11a uses coefficients found using monthly mean Chl-a and $R_{RS}$. And yes, the different functions are given by light blue and black lines in Figure 4. We plan to add a further sentence to Fig 11 to make this point. Thus together with the sentences already in the caption we will have the following to clarify this different (underlined are altered text):

*"In these plots, monthly mean output of Chl-a and $R_{RS}$ were used to calculate the algorithm, and only monthly mean output is shown (4 million versus 140 million points), thus at a great computational savings. The difference in the algorithm is shown in Figure 4 (the light blue line is the algorithm with coefficients found using daily values, versus the solid black line where coefficients where found using monthly values). Differences between 11a and 5a are due to this difference in sampling (discussed in Appendix B). Also notice the difference in values on the colourbars between this figure and Figure 5."*

P12 L20-21 & L33-P13 L1: I'm not so sure it is quite as simple as this. I agree that the other optically significant materials are contributing to false Chl-a signals: there is a shoulder in all the derived Chl-a time series (Fig 8a), that aligns perfectly with the peak in CDOM and detritus (Fig 8b). But you can see the pattern of the actual Chl-a signal in the derived values, with peaks aligning on around days 60 and 75 - the magnitude of these derived values are just less than the actual values, but I'd say these are the "true peaks" of the spring bloom. After approximately day 75, there is the interference from CDOM and detritus, hence when calculating the initiation of the spring bloom (as described in the appendix), this large "false peak/shoulder" increases the median Chl-a and skews the determined initiation date. So I think what your data could be showing is (1) the Chl-a products do capture the peak of the spring bloom, but the magnitude of that peak is too small, and (2) the CDOM and detritus contribute to a false Chl-a signal, which makes it appear as if the bloom lasts longer and (depending on how you define bloom initiation) makes the initiation date appear to lag compared to the model actual.

We agree that that there is an alignment with the first peak (around day 60), but there is an offset for the maximum peak. Indeed, some of the points we make here are subject to what we define as the "initiation" and "peak", and we now make this clearer in the revised text: see below.

We use the definitions of Cole et al (2012) and to make our points clearer, we have added extra lines to Fig 8 (see below) to show the timing of the "initiation". Thus, for this definition of initiation there is indeed a lag between the derived products and the actual Chl-a. (Fig 9)
We add text to clarify this – i.e. acknowledging that the products are capturing the peaks at some times and not others. We also add a figure of the maximum peak offset as an additional panel for Figure 9 (b), as well as for the difference in timings of initiation for CDOM and detrital matter in response to a comment by reviewer 2.

Section 4 (pg 9 and 10) will now read (underlined indicates new or altered text):
*"We have noted that in all approaches, though even more obvious in RA, there is a seasonally altering pattern between the derived and actual model Chl-a (Fig 6). The amplitude of the peak of spring blooms is often underestimated in the products derived using global coefficients (GS and GA) in high latitude, especially in the subsampled algorithm (GS) (Figs 6). , Derived Chl-a values were also often higher than model actual Chl-a outside of bloom peaks. We consider the phenology using a single location (in the subpolar North Atlantic) for a single year as illustration (Fig 8a). Though the derived products show similar (though smaller) peaks to the actual Chl-a, and sometimes similar peak timing early in the season (see for instance the first distinct peak in this illustrative location), there are noticeable lags for the maximum peak (shown with a dotted line) and other mismatches later in the season. We also find that the bloom period lasts later into the year. The actual Chl-a also starts its sharp increase in spring (the initiation of the spring bloom, shown with dashed line) considerably before all three derived products (Fig 8a). We follow the approach of Cole et al (2012) for determining the "initiation of the spring bloom" as the time when the Chl-a first increases 5% above the annual median (horizontal dashed line, more description in Appendix A).*

*Figure 8 shows just one location for 1 year. To consider the large scale patterns, we determine the lag in the spring initiation (Fig 9a) and maximum bloom timing (Fig 9b) for each location averaged over all years. We find that in almost all locations the derived Chl-a shows the bloom starting later than the model actual Chl-a (Fig 9a). This offset is typically by about 5-10 days but can be as much as 30 days. The maximum Chl-a from the derived product also lags the actual Chl-a in most locations, though by only a few days (Fig 9b). These results indicate that temporal as well as spatial biases occur as a result of deriving Chl-a from X and suggests care should be taken when calculating phenology from satellite products or when evaluating phenology in models using satellite-derived Chl-a. We discuss the reason for the lags in the next section."*

And the following caveats in section 5 (pg 11, lines 5-11):
*"We add the caveats that the exact definition of "initiation of bloom" does impact how much of a lag there is in the phenology. For instance, if the first peak in the model actual Chl-a in Figure 8a was defined as "the spring bloom" we would suggest the derived Chl-a does capture the timing better (though not the magnitude). We also note that the model parameterization of CDOM and detrital particle are not necessarily sufficiently well developed to make quantitative statements on the likely real-world lags. Thus, though we do suggest there could be significant lags in phenology in the real world satellite Chl-a product, we do not suggest that the values in Figure 9 are necessarily accurate for the real world. This analysis should instead be seen as a cautionary statement about using satellite-derived products for phenology of the quantities for which they are proxies."*

And more-over are careful in the rest of the text to discuss "phenology" rather than "spring bloom" such that the role of definitions of timing are not as relevant.

New versions of Figure 8 and 9 and their captions (altered of added text underline) are provide here. Extra panels in Figure 9 are added at Reviewer 2's suggestion.

[Figure]

***Figure 8:*** *Illustrative example timeseries for one year from a single location in the North Atlantic (shown as x on Fig 9). (a) "actual" Chl-a (black), derived Chl-a using subsampled output (GS, light blue), derived Chl-a using all output (GA, dark blue), and the Chl-a product derived using a regional specific algorithm (RA, purple). (b) actual Chl-a (black), CDOM (red) and detritus (green), all normalized to their peak value. Dashed vertical line indicates the "initiation of the bloom" which is taken to be when Chl-a reaches 5% above the annual median value following Cole et al (2012) and discussed further in Appendix A (dotted horizontal line shows this value for the model actual Chl-a). The vertical dotted line indicates the peak of the bloom. Shown here is only a single year and location, however for larger scale perspective, the difference in initiation and peak timing between model actual and derived Chl-a averaged over all years are shown for the globe in Figure 9.*

[Figure]

*Figure 9: Lag in phenology. Number of days between a) the initiation of the spring bloom from model actual Chl-a and that for the model derived Chl-a (GS); b) yearly maximum of model actual Chl-a and that for the derived Chl-a (GS); c) initiation of the spring bloom from model actual Chl-a and the initiation of the CDOM increase; d) initiation of the spring bloom from model actual Chl-a and the initiation of detrital particle increase. Bloom initiation is defined as when Chl-a, CDOM or detrital particles reach 5% above their annual median value (see Appendix A). White areas indicate regions with no significant seasonal cycle or are not resolved by the model (e.g. Arctic Ocean).*

It might be useful to have a table presenting the numerical results (e.g. log/linear RMSE, absolute % bias, etc.) for each approach in Section 3 and 5, to make it easier to compare the different results.

We now plan to add two tables (Table 2 and 3, shown below), one for each section. We split into two tables, to avoid comparison between the daily and the monthly determined algorithms. We also emphasis this difference in the table 3's caption

|  | Approach 1: GS | Approach 2: GA | Approach 3: RA |
|---|---|---|---|
| $r^2$ (log space) | 0.9088 | 0.9222 | 0.9466 |
| RMSE (log space) | 0.1599 | 0.1477 | 0.1215 |
| $r^2$ (linear space) | 0.6014 | 0.7670 | 0.8301 |
| RMSE (linear space) | 0.4816 | 0.3682 | 0.3083 |
| absolute % bias | 22% | 23% | 17% |

*Table 2: Results of comparison between model "actual" and model "satellite-like" derived Chl-a for the three algorithm approaches discussed in Section 3. Statistics are calculated for each grid and each day*

|  | Default | EXP-1:
uniform $a_{CDOM}$ | EXP-2:
uniform $a_{det}$ | EXP-3:
phytoplankton optical same |
|---|---|---|---|---|
| $r^2$ (log space) | 0.8999 | 0.8742 | 0.8905 | 0.9493 |
| RMSE (log space) | 0.1678 | 0.1636 | 0.1663 | 0.1208 |
| $r^2$ (linear space) | 0.5373 | 0.6298 | 0.5991 | 0.7520 |
| RMSE (linear space) | 0.4420 | 0.3811 | 0.3962 | 0.2591 |
| absolute % bias | 21% | 20% | 23% | 18% |

*Table 3: Results of comparison between model "actual" and model "satellite-like" derived Chl-a for the sensitivity experiments discussed in Section 5. All "satellite-like" derived Chl-a was calculated using the GS approach. "Default" is the full experiment discussed in Section 3, but with monthly $R_{RS}$ used to calculate the algorithm coefficients. Statistics are calculated for each grid cell and each month over 15 years, except for grid cells and times with low light, very low Chl-a and shallow regions (see text).*

Fig 4 and Fig 8: Legends would make these plots easier to read.

Agreed. We add these in the revised paper (see version of Fig 8 above).

Fig 5 and Fig 11: it would be useful to have a title or text on each graph to show which subplot is which e.g. "(a) GS"

Agreed, we add these in the revised paper.

Fig 8: The thick black line on top of the other time series signals masks some of the detail of the derived Chl-a products, particularly after the first 3 months - could this be represented in a different way? Also, check the axis labels, I think Fig 8b is showing days, not months.

We've redone the figure with a thinner black line, placed under the other lines. The resulting figures is clearer (see above). And yes, Fig 8b x-axis was days, we now change so that Fig 8a and 8b have the same x-axis. Thank you for catching this.

Technical Corrections:
P1 L15-16: missing the word "to" i.e. sentence should read "...derived Chl-a to the actual..."

Thank you – added in the revised text.

P1 L25: should be either "These results indicate" or "This result indicates"

Thanks –fixed in the revised text as "These results indicate…."

P9 L29: Should this sentence not end with a question mark? i.e. ". . .community structure)?"

Yes, fixed in revised text.

P11 L23: remove the second "like"

Thanks, done.

P12 L24: build should be built

Yes, thank you. Will fix in revised text.

We thank the reviewer for their comments, and respond to each point below in blue text, and proposed altered text in italics.

The paper 'Modelling Ocean Colour Derived Chlorophyll-a' by Dutkiewicz et al. uses a modified version of the MIT general circulation model that incorporates scattering and absorption properties of water, detritus, coloured dissolved organic matter (CDOM) and 9 phytoplankton types. 'Actual' chlorophyll-a concentration (Chl-a) is determined by summing the variable chlorophyll-a concentration of the 9 phytoplankton types. Remote sensing spectral reflectance, determined from the resulting modelled upwelling and downwelling irradiances are used in a NASA OCx type algorithm to compute satellite-like 'derived' chlorophyll-a concentration which is compared to 'actual' chlorophyll-a concentration. Firstly, the model output is used to test assumptions used in the derivation of chlorophyll-a concentration using the standard OCx ratio algorithm. Secondly, bloom initiation timings, determined from the 'derived' and 'actual' chlorophyll concenrations, are compared. Lastly, the impact that other optically important parameters may have on 'derived' chlorophyll concentration is explored. The authors conclude that (a) applying a single set of coefficients in the OCx algorithm globally is not as accurate as applying regionally variable coefficients, (b) there is a temporal mismatch between the initiation of the spring bloom defined using 'actual' Chl-a compared with that de-fined using 'derived' Chl-a and (c) that this mismatch may be caused by the optical influence of other substances such as CDOM and/or detritus.

I found this paper to be generally well-thought out and well-written. For colleagues who are not regular users of these products, this paper provides important caveats for the use of satellite derived data. For those of us who work with ocean colour derived products more often, the results may not be unsurprising, but it serves as a timely reminder of the limitations associated with satellite derived data.

We thank the reviewer for these positive comments and for recognizing that though some of the results are not surprising to those who are experts in ocean colour products, these results offer a reminder of the limitations. We do believe that for non-experts of ocean colour products, these results will be very informative.

Whilst interesting and worthy of publishing in Biogeosciences, I have some concerns with some of the quite sweeping conclusions that appear to be derived from comparison with a single dataset or single datapoint (referred to in more detail below). In addition, I have made other specific comments that I believe should be addressed be-fore this manuscript is suitable for publication.

The reviewer's comment relates to Fig 8, and the discussion of the results shown in it for one location and one year. This figure was only supposed to be illustrative, not the main basis for our conclusions. We have significantly rewritten this section to make this obvious, and include several more panels in Fig 9 that show that the inferences we gain from that location are relevant for much of the globe. See below.

SPECIFIC COMMENTS

P 1, L27 – I am unclear whether the term '. . .real world Chl-a. . .' used here refers to actual in-situ chlorophyll-a concentration or satellite derived chlorophyll-a concentration. The term 'real' appears to be used interchangeably throughout the manuscript.

We agree that this incident of the use of "real" was confusing and will change this to be "to real world satellite-derived Chl". We have also gone through the manuscript being careful how we use "real world" in the text. We also add a sentence in the introduction (pg 3, line 27-29) to define "real world" for the rest of the paper:

*"(In this article "real-world" will be used to refer to the real ocean and the real derived ocean colour products that are provide by space agencies. The "real world" is thus different to the numerical biogeochemical/ecosystem/optical model output and the products derived from it.)"*

P5, L15 – Would it make more sense to swap Figures 1 and 2 so that the comparison of model and OC-CCI reflectance appears first as Fig 1 and is followed by the product comparisons as Fig 2. If this were the case, would there then be anything gained by comparing model actual and derived Chl-a to the OC-CCI Chl-a product? As I under-stand it, this paper is more about using the model output as a test ground to compare model 'actual' to 'derived' Chl-a rather than testing how well the model replicates the real world' values. Just a thought.

We like the "thought" – it would make a cleaner paper not getting into the aspect of evaluating the model. However, one of the points we would like to make is that comparing model "actual" or model "derived" Chl-a to "real-world" satellite derived Chl-a leads to different results. There are therefore important implications of our analysis for model validation (i.e. that comparing model chl-a to real-world satellite-derived Chla is not comparing like-for-like), so we feel this comparison is an important part of the paper. We have considerably rewritten the relevant paragraph (last one of section 3.1, pg 7, lines 17-31) to make this point clearer and to emphasis that this is not a "evaluation", but a demonstration (underlined is added or altered text).

*"Finally in this section, we ask: Which model Chl-a (derived versus actual) best matches real-world OC-CCI product? We do not do this not for model validation purposes (see evaluation in Dutkiewicz et al., 2015), but rather to re-emphasis that the satellite derived Chl-a products are proxies for real world actual Chl-a: The two are not the same thing. We compare climatological monthly model derived Chl-a and model actual Chl-a to OC-CCI monthly climatology regridded to the model configuration (1 degree resolution). We find that the model derived Chl-a has global RMSE of 0.2867mg/m$^3$, which is significantly lower than 0.6370mg/m$^3$ found when comparing model actual Chl-a to OC-CCI. Comparisons are particularly better for the Southern Ocean and North Pacific (Fig 1). Consequently, some (though certainly not all) of the biases noted when comparing model actual Chl-a (Fig 1b,e) to real world satellite derived Chl-a products (Fig 1a,d, section 2 and in the model evaluation done in Dutkiewicz et al. 2015) are due to the real world Chl-a derived product bias and not a deficiency in the biogeochemical/ecosystem/optical model. It follows that a model satellite-like derived products (Fig 1 c,f) might be a better evaluation tool for comparing to ocean colour products derived with the same algorithm (Fig 1a,d) than the model actual Chl-a fields themselves."*

Though we agree that it would be nice to have figure 2 (reflectance) before figure 1 (product: Chl-a) we could not find a way to do this without complicating the paper. So we have left as is.

P6, L13 – The authors talk about comparing '. . .locations and dates similar to those in NOMAD.' What is their definition of similar?

The model has 1 degree resolution – so we are using the degree box within which the actual is situ measurements are taken. This is unnecessarily confusing though and we now state differently (pg 6, line 27):

*".. at locations and dates nearest in time and space to those in NOMAD"*

P6, L15 – Again they use 'similar' to describe the resulting relationship between model 'actual' chlorophyll, model X, and real world in situ observations without really defining what similar means.

Here we had meant to reference Figure 4, where we show the OC4 and OC3M-547 functions. We now change this sentence (pg 6, line 29) to reflect this:

*"The resulting relationship between model blue/green reflectance ratio (X) and Chl-a from subsampling the model (Fig 3a) is similar to that found for real-world algorithms (Fig 4)."*

P6, L20 – The authors make no mention of the discrepancy between model reflectance wavebands (blue – 450 nm, 475 nm or 500 nm, green – 550 nm) and those used in the OC4 (blue – 443 nm, 490 nm or 510 nm, green – 555 nm) or OC3M-547 (blue – 443 nm or 488 nm, green – 547 nm) algorithms when comparing coefficients in Table 1. It might make it clearer to those not familiar with the derivation of these algorithms that they are not comparing like with like.

Yes, thank you. This is an important point and in the revised version we will add the following to the text and to the figure caption.

We will add where we compare the model RRS to OC-CCI Rrs (pg 5, line 27-29):

*"We note that the model does not have the exact same wavebands as any of the ocean colour satellites, and as such here we compare to the nearest bands: 450nm model to 443nm for the OC-CCI product, and 550nm model to the 555nm OC-CCI product."*

Also when we compare the model GS coefficients to OC4 and OC3-457 (Fig 4 and Table 1) we now add the following sentence (pg 6, line 30 to pg 7, line 1):

*"Some of the differences between real-world and model coefficients is likely to come from the use of different exact bands in the blue and green (e.g. 550nm for model green versus 555nm for OC-CCI)."*

We also add the following to the figure caption in Figure 2:

*"We compare the model wavebands against the nearest OC-CCI wavebands, but note that they are not identical."*

P7, L1-5 – The authors compare the OC-CCI Chl-a product to model derived Chl-a (although see my second comment). Where are the plots to support the statistics? What monthly climatologies are used to generate these statistics (is it a combination of Jan and Jul or all months?) Over what period are the January and July OC-CCI mean values determined? Are the OC-CCI output averaged to 1 degree by 1 degree similar to the model output? What version of the OC-CCI product is used? Are these OC-CCI products just OCx type output or do they include data from the Hu CI algorithm? The OC-CCI output is just one product. The statement at L5 seems to be quite a bold statement to make when only one product has been compared.

The product from OC-CCI was OC4 without the Hu CI component. We discuss the Hu et al adjustments now in the introduction and text (see comments by Reviewer 1).

The RMSE is determined from daily values over the full time period (not just Jan and Jul) averaged globally. We have regridded OC-CCI onto the same grid as the model for these comparisons.

We would prefer not to add any additional figures, especially for this point. However, we do agree with the reviewers' concern on this section and have considerably rewritten it. We add some additional text (also in the figure caption) to explain the statistics better and state what OC-CCI product we are using. Additionally we add text in both introduction and summary to highlight that OC4 is just one product, and that there are newer (and better) products also available. We have stuck here to the OC4 as it was the one that compared to the version of OC-CCI that we downloaded (a new version (V3) has just been released). We do specifically mention the newer OCx+CI versions. We are not convinced that the statement on L5 is too bold. However we do add a caveat in it.

The paragraph questioned here is provided above for the discussion on Fig 1 above. And we add the following sentence to caption of Figure 1:

*"We use version 2 of the OC-CCI, which uses an OC4 algorithm for determining the Chl-a product, and thus comparable algorithm as used in our model derived Chl-a shown in e,f."*

In the introduction (pg 3, line 19-27) we have the following (underlined is added or altered text):

*"There is significant ongoing work to improve algorithms. For instance, the newest National Aeronautics and Space Administration (NASA) reprocessing of Chl-a products has included a merged approach that uses different combination of reflectance bands at low Chl-a (Hu et al., 2012). There have also been many attempts to develop more mechanistically derived algorithms (e.g. using known relationships between absorption, scattering and reflectance). Here we focus on the Chl-a estimated from the blue/green reflectance as it is still the most commonly known product, and until very recently used in products downloaded from both NASA and the European Space Agency (ESA) data portals, as well as merged products such as the Ocean Colour Climate Change Initiative (OC-CCI). However we note that similar techniques used in this paper could help inform on other algorithms."*

The final sentence of the summary (pg 15, lines 27-30) now reads (underlined is new text):

*"We also hope that the ocean colour community will see the potential of model approaches such as this for deriving sampling strategies, further studies on newer Chl-a algorithms (e.g. NASA Reprocessing 2014.0, and OC-CCI V3 release), other ocean colour products, and will help with algorithm developments for current and future ocean colour measurements."*

P7, L20 – Perhaps it's my eyesight but I'm not convinced that Figs (b) and (e) show '. . .much lower biases at high latitudes. . .'. (I assume you are comparing Fig 6 (b) and (e) to Fig. 6 (a) and (d))

Yes we were comparing b,e to a,d – will make clearer in the revised version. And we agree that this is not obvious from the figure – the improvement is mostly just at the edge of the data before moving into the white areas. We now restate this sentence (pg 8, line 15-16, underlined is new/altered text):

*"Though there is some improvements in some regions in the higher latitudes, there is actually decrease in skill at lower latitudes (Fig 6b,e compared to a,d). There is in fact a slight increase in the mean % absolute bias (23%) between this and the GS estimates: When transformed into percent errors the increased biases at low Chl-a, low latitude regions become more prominent."*

P8, L3-5 – Are grid cells with depths less than 1000m also excluded?

Yes, will be stated in the revised version.

P8, L22-24 – These statements appear to be derived from data taken from one point in the North Atlantic. Is this a fair representation of the global pattern or is it just representative of this location?

We use the one location as an illustration. The results shown in figure 9 show that same lag in initiation of the bloom occur across much of the high latitudes. We make clear in the text and Figure 8 caption that this is just an illustration. To further strengthen the relevance of this discussion in space and time, we now include other panels in figure 9 to show the mismatch in the peak timing, as well as in the initiation of increase in CDOM and detrital matter relative to actual Chl-a.

Section 4 (pg 9 and 10) now reads (underlined is altered or added text):

*"We have noted that in all approaches, though even more obvious in RA, there is a seasonally altering pattern between the derived and actual model Chl-a (Fig 6). The amplitude of the peak of spring blooms is often underestimated in the products derived using global coefficients (GS and GA) in high latitude, especially in the subsampled algorithm (GS) (Figs 6). , Derived Chl-a values were also often higher than model actual Chl-a outside of bloom peaks. We consider the phenology using a single location (in the subpolar North Atlantic) for a single year as illustration (Fig 8a). Though the derived products show similar (though smaller) peaks to the actual Chl-a, and sometimes similar peak timing early in the season*

[revised manuscript text omitted]

P9, L4 – The authors could reference the Dutkiewicz et al. (2015) paper again here.

Good idea, done in the revised version.

P9, L5 – The authors refer to 'studies' then reference a single instance.

We add "e.g" to the text and additional references (e.g. Loisel et al., 2010; Brown et al., 2008, Siegel et al., 2005a, 2005b)

Brown, C.A., Huot, Y., Werdell, P.J., Gentili, B., and Claustre, H.: The origin and global distribution of second order variability in satellite ocean color and its potential applications to algorithm development, Remote Sensing of Environment, 112, 4186-4203.

Loisel, H., Lebac, B., Dessailly, D., Duforet-Gaurier, L., and Vantrpotte, V.: Effect of inherent optical properties variability on chlorophyll retrieval from ocean colr remote sensing: an in situ approach. Optics Express, 18,

Siegel, D.A., Maritorena, S. Nelson, N.B., and Behrenfeld, M.J.: Independence and interdependencies of global ocean color properties; Reassessing the bio-optical assumption. J. Geophys. Res., 110, C07011, doi:10.1029/2004JC002527, 2005a

Siegel, D.A., Maritorena, S., Nelson, N.B., Behrenfeld, M.J. and McClain, C.R.: Colored dissolved organic matter and its influence on the satellite-based characterization of the ocean biosphere, Geophys. Res. Letters, 32, L20605, doi:10.1029/2005GL024310, 2005b.

P9, L17-21 How do the authors support this statement? If it is the timeseries data in Figure 8, then these are data from just one point in the North Atlantic. I don't think that data from one location and for one year is sufficient to warrant these conclusions.

See our comments above for P8, L22-24 above. The results in Figure 9 show the global results averaged over 13 years. However we do see the reviewers point, and now add the extra panels (c,d) to Figure 9. We modify the text in Section 5 (original P9, L17-19) as such (underlined is added or altered text):

*"However, we find that though linked, there are noticeable lags in the sharp increase in accumulation (Fig 8b, Fig 9 c,d) and peak timing and decline(Fig 8b) between CDOM and detrital matter and the model actual Chl-a."*

Figure 3 – How do you differentiate between zero bias and lack of data? Could I suggest that lack of data is coloured differently to zero bias?

I assume you mean Figure 6? Good idea. We will do have done so in the revised paper.

Figure 4 – Not sure whether the figure order works. The first mention of Fig. 4 that I can find occurs on P11 after reference to all the other figures. Again, if the authors are comparing the polynomials it might make it clearer to the reader if they acknowledge that different wavelengths have been used in the derivation in the legend.

Figure 4 is mentioned first on pg 6 (line 22), but we now also reference it earlier in the newer version of the paper. Since two of lines in Figure 4 are those in Figure 3, it does quite naturally belong here.

We add the acknowledgment of different wavelengths in the caption of Figure 4:
*"Note that the algorithms for the model come from band ratio of 425nm/450nm/475nm and 550nm. For the real world algorithms the band ratios are different and specific for the satellite sensor (SeaWifs or MODIS)."*

Figure 8 – I don't think the x axis matches the label in Fig 8 (b). I assume the vertical dotted line marks the peak in 'actual' Chl-a?

Thank you for catching this – we have altered the axis to match Fig 8a. Yes, the vertical dotted line is the peak – we now mention this in the figure caption. We have also added a line to show the initiation of the spring bloom for clarity. See new figure 8 and caption above.

TECHNICAL CORRECTIONS

P1, L16 – Should read '. . .Chl-a to the actual. . .'

Thank you – we fixed this.

P1, L25 – Should read 'This result indicates. . .'

Actually, I think should read "These results indicate…", so will change to this instead. Thank you for catching this inconsistency.

P2, L15 – I think this is the first use of the acronym CDOM and so it should be defined here.

Yes, added.

P2, L18 – Should read 'There have been. . .'

Yes, thank you.

P2, L20 – Should read 'product' instead of 'products'

Yes, fixed

P2, L24 repeats L7

We removed the text at L24 from the revised text.

P3, L28 – In situ is italicised here but nowhere else.

Thanks – we removed the italics to be consistent.

P4, L14, 15, 17, 19 - Repeated uses of 'explicit'.

Yes, have removed excessive use of "explicit". Thank you.

P5, L15 – 'Fig.2' is italicized

We removed the italics. Thank you.

P7, L4 – Missing figure number

Corrected.

P7, L19 - Should read '. . .lead to a better. . .'

Yes, thank you.

P9, L3 – Should read 'lead' rather than 'leads'

Yes, thank you

P9, L21 – Should read '. . .remains relatively high. . .'

Yes, fixed

P9, L29 – Don't think there should be a comma after 'pigments'.

Indeed not, we have removed.

P9, L31 – However, I think there should be one after 'reflectance'.

Yes, have fixed this in revised version.

P12, L22 Should read '. . .by-products. . .'

Yes, thank you.

[revised manuscript text omitted]

---

## Author Response (AR2)

Dear Emmanuel –

Thank you for the additional comments on our paper. I respond here to your comments – line and page numbers match the revised version attached to this document.

Pg 1, line 20: Yes, it is not surprising that the regional algorithms work better over the course of the full year, and we now add text to make this obvious. However what did surprise me is that there are times of year when the results are worse – this is made clearer later on in the text.

Pg 2, line 11: Yes, Morel and Prieur (1977) is a good reference here, we have added it.

Pg 2, line 24: Yes, Werdell et al (2013a) is a good reference here – we have added it.

Pg 2, line 32: We have changed to "biogeochemical and ecosystem" models as suggested

Pg 3, line 26: Yes, the second "real" is superfluous – we have removed.

Pg 4, line 30 (and elsewhere): No, we do not take into account salinity. We also do not take into account viruses and mineral components. We now add a sentence to make this apparent:" It does not however include additionally potentially important components such as minerals and viruses (Stramski et al. 2001) or salt (Werdell et al., 2013b)." We agree that this might help in the low Chl-a regions, and indeed something to include in a newer iteration of the model.

Pg 7, line 10 (and elsewhere, including Tables 1 and 2 and Figures 5 and 11): We have reduced to only two significant digits.

Pg 7, line 17: We have restructured the sentence as suggested.

Pg 7, line 31: Agreed that the RT model is still simple. But even so we maintain that a better comparison is model remotely sensed Chl to real world remotely sensed Chl. We believe that adding any qualifier here would detract from this important message. As such we keep as is.

Pg 8, line 33: Indeed it has to be better – and we now add "Unsurprisingly..". However these is still significant seasonal bias and times of year when the solution is even worse. This was surprising to me at first.

Pg 9, line 11: Here you comment that the log-nature of the fitting means that in absolute sense the higher Chl values should has larger uncertainty. We agree with this, and add a statement to this effect on pg 7, line 13: "Larger errors at higher absolute concentrations are anticipated given the polynomial fitting was done in log space." However we do not believe this is the reason for underestimate of the peak's of bloom. We agree that the high values should be captured worse, but not always under-estimated. So we do not add any additional statement here.

Pg 9 line 20 (and elsewhere): Comments on the metrics used are (as you later realized) dealt with in the discussion. As such we do not make any changes here.

Pg 10, line 26: We add "in the deep ocean" here.

Pg 13, line 24: We add the Haentjens et al., 2017 reference here.

Pg 15, line 24: We have changed to "biogeochemical and ecosystem" models as suggested

Pg 16, line 11: This was for daily mean value. We have added "daily mean" to make this clear here.

We have altered Table 1 and 2 and redone Figure 5 and 11 as well as added 4 additional references in response to these comments.

Thank you for your help with this paper!

Yours

Stephanie

[revised manuscript text omitted]